# Unifying Text Semantics and Graph Structures for Temporal Text-attributed Graphs with Large Language Models

**Siwei Zhang[1], Yun Xiong[1]\*, Yateng Tang[3], Jiarong Xu[2], Xi Chen[1]**
**Zehao Gu[1], Xuehao Zheng[3], Zi'an Jia[1], Jiawei Zhang[4]**
[1]Shanghai Key Laboratory of Data Science, School of Computer Science, Fudan University
[2]School of Management, Fudan University [3]Tencent Weixin Group
[4]IFM Lab, University of California, Davis
{swzhang24, x_chen21, guzh22, 24110240035}@m.fudan.edu.cn   jiawei@ifmlab.org
{yunx, jiarongxu}@fudan.edu.cn   {fredyttang, xuehaozheng}@tencent.com

## Abstract

Temporal graph neural networks (TGNNs) have shown remarkable performance in temporal graph modeling. However, real-world temporal graphs often possess rich textual information, giving rise to temporal text-attributed graphs (TTAGs). Such combination of dynamic text semantics and evolving graph structures introduces heightened complexity. Existing TGNNs embed texts statically and rely heavily on encoding mechanisms that biasedly prioritize structural information, overlooking the temporal evolution of text semantics and the essential interplay between semantics and structures for synergistic reinforcement. To tackle these issues, we present CROSS, a flexible framework that seamlessly extends existing TGNNs for TTAG modeling. CROSS is designed by decomposing the TTAG modeling process into two phases: (i) temporal semantics extraction; and (ii) semantic-structural information unification. The key idea is to advance the large language models (LLMs) to *dynamically* extract the temporal semantics in text space and then generate *cohesive* representations unifying both semantics and structures. Specifically, we propose a Temporal Semantics Extractor in the CROSS framework, which empowers LLMs to offer the dynamic semantic understanding of node's evolving contexts of textual neighborhoods, facilitating semantic dynamics. Subsequently, we introduce the Semantic-structural Co-encoder, which collaborates with the above Extractor for synthesizing illuminating representations by jointly considering both semantic and structural information while encouraging their mutual reinforcement. Extensive experiments show that CROSS achieves state-of-the-art results on four public datasets and one industrial dataset, with 24.7% absolute MRR gain on average in temporal link prediction and 3.7% AUC gain in node classification of industrial application.

## 1 Introduction

Temporal graphs are crucial for modeling dynamic interaction data, where objects are represented as nodes and timestamped interactions are depicted as edges [1, 2]. Unlike static graphs, temporal graphs continuously evolve over time [3]. To capture the temporal dependencies and realize representation learning for such graphs, extensive research has developed temporal graph neural networks (TGNNs) [4, 5, 6, 7, 8]. These works typically adopt structural encoding mechanisms [9, 10] to encapsulate the dynamics of graph structures [11], thus enabling representations for downstream tasks.

Meanwhile, besides the dynamically evolving graph structures, real-world temporal graphs are also often accompanied by rich text attributes, giving rise to temporal text-attributed graphs (TTAGs) [12]. As shown in Fig. 1(a), in e-commerce networks, nodes may encompass elaborate text descriptions such as user profile or product introduction, while timestamped edges can attach transaction details

---

*Corresponding Author.

39th Conference on Neural Information Processing Systems (NeurIPS 2025).

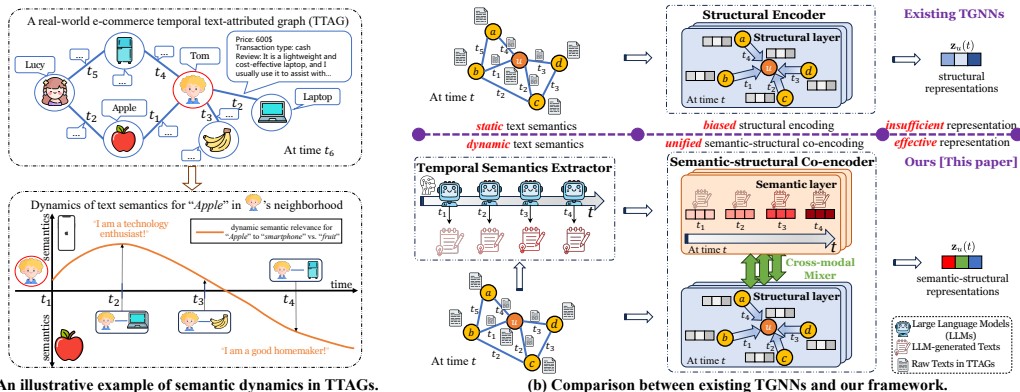

Figure 1: **(a) An illustrative example of semantic dynamics in TTAGs.** Temporal text-attributed graphs (TTAGs) inherently exhibit semantic dynamics, where text semantics around nodes dynamically evolve across temporal dimensions. **(b) Comparison between existing TGNNs and our framework.** Going beyond existing TGNNs that biasedly focus on structural dynamics, our framework seamlessly unifies text semantics and graph structures of TTAGs, promoting both dynamic semantic understanding and semantic-structural reinforcement.

including price, transaction type, review, *etc.* Representation learning in TTAGs presents unique challenges due to the intricate combination of dynamic text semantics and evolving graph structures, which remains largely under-explored in the community. Despite the success of existing TGNNs, they still encounter two fundamental limitations that hinder their accommodation to TTAG modeling.

**Limitation (i): Neglect of semantic dynamics.** Text semantics arise from the textual attributes of a node and its surrounding neighborhoods. These semantics exhibit changes across timestamps due to the temporal evolution of neighborhood contexts within TTAGs. For example, as depicted in Fig. 1(a), the term "*apple*" may take on different meanings according to user preferences (driven by its neighborhoods), such as "*smartphone*" when focused on technology while "*fruit*" when turning to daily life. Such characteristics necessitate a temporal-aware design for extracting text semantics over time. However, existing TGNNs always use pre-trained language models, *e.g.*, MiniLM [13], to statically embed texts as pre-processed features, failing to adapt to such dynamic semantic shifts and leading to suboptimal representations for effectively capturing the semantic dynamics.

**Limitation (ii): Ineffective encoding for semantic-structural reinforcement.** Another limitation of existing TGNNs is the rigid reliance on their structural encoding mechanisms, which predominantly focus on topological information without adequately incorporating semantic considerations. We argue that the semantics and structures of TTAGs can mutually reinforce each other, making a biased encoding mechanism that solely emphasizes structures untenable. This hypothesis is reasonable due to the inherent complementarity between textual content and structural connectivity [14, 15] (also empirically validated in Sec. A). For instance, product recommendations are shaped not only by goods' descriptions but also by users' historical behaviors [16]. Existing TGNN encoding mechanisms do not account for such nuanced semantic-structural reinforcement in TTAGs, resulting in insufficient representations that are overly reliant on simplistic (or sometimes noisy) structural information.

On the other hand, recent surveys [17, 18] reveal that large language models (LLMs), *e.g.*, DeepSeek-v2/3 [19], exhibit notable capabilities in semantic understanding and generation, which have been successfully leveraged in graph modeling through text augmentation [20, 21]. However, LLMs employed in these methods are typically limited by static corpora input for reasoning, making them ill-suited to capture the fine-grained dynamics of TTAGs. This limitation inspires us to ask: *Can we advance LLMs for TTAG modeling as a promising solution to the aforementioned limitations of existing TGNNs?*

**Key innovation.** Beyond existing TGNNs that solely prioritize structural dynamics, in this paper, we tackle above issues by decoupling the TTAG modeling process into two phases: (i) temporal semantics extraction; and (ii) semantic-structural information unification. By extracting both dynamic semantics and evolving structures of TTAGs, we cohesively unify them to harness their dual-strengths for more informative representations. Consequently, as shown in Fig. 1(b), the key idea of this work is to elevate LLMs with dynamic reasoning capability to extract semantic dynamics, which subsequently allows for a unified, integrated, and non-biased encoding mechanism obeying semantic-structural reinforcement.

**Present work.** We extend existing TGNNs for TTAG modeling and introduce CROSS (Cohesive Representations Of Semantics and Structures), a novel framework that seamlessly unifies text

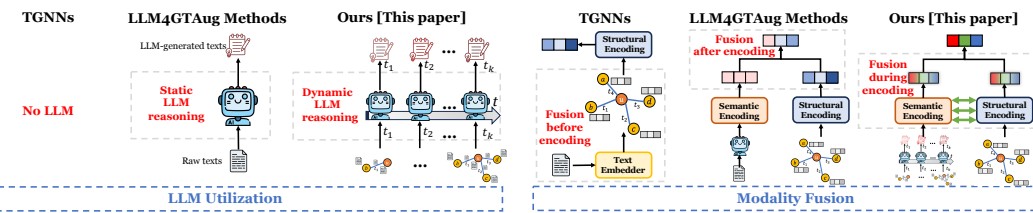

Figure 2: **Technical difference comparison from the perspective of LLM utilization and modality fusion.**

semantics and graph structures coupled with LLMs. CROSS can significantly boost existing TGNNs, and it comprises two main components: (i) Temporal Semantics Extractor, and (ii) Semantic-structural Co-encoder. To capture the semantic dynamics for nodes within TTAGs, we propose a Temporal Semantics Extractor, which empowers the LLMs with the dynamic semantic summarization reasoning capability for TTAG modeling. It automatically detects semantic dynamics by constructing a temporal reasoning chain to dynamically prompt the LLMs to summarize the textualized, evolving neighborhoods of nodes, thus revealing the linguistic nuances across temporal dimensions. Besides the Extractor, to effectively achieve semantic-structural reinforcement, we further propose the Semantic-structural Co-encoder that simultaneously exploits both semantic and structural information through an iterative, multi-layer design. Each layer in the proposed Co-encoder bidirectionally transfers the cross-modal information between semantics and structures, allowing both modalities to blend deeply and fully illuminate each other. By performing such a unification, we can generate modal-cohesive representations that are both semantic-enriched and structural-informed.

The main contributions of this paper are summarized as follows:

- We address an under-explored problem of temporal text-attributed graph (TTAG) modeling and propose CROSS. To the best of our knowledge, CROSS is the first framework designed to unify text semantics and graph structures with LLMs for TTAG modeling.
- We design *a temporal-aware LLM prompting paradigm* for TTAG modeling and develop Temporal Semantics Extractor. It enhances LLMs with dynamic reasoning capability to offer the evolving semantics of nodes' neighborhoods, effectively detecting semantic dynamics.
- We introduce *a modal-cohesive co-encoding architecture* for TTAG modeling and propose Semantic-structural Co-encoder, which jointly propagates semantic and structural information, facilitating mutual reinforcement between both modalities.
- We conduct extensive experiments on four public datasets and one practical e-commerce industrial dataset. CROSS outperforms baselines with 24.7% absolute MRR gain on average in temporal link prediction and 3.7% AUC gain in node classification of industrial application.

## 2 Preliminaries and Related Work

### 2.1 Preliminaries

*Definition 1.* **Temporal Text-attributed Graph.** Given a set of nodes $\mathcal{V}$, node text attributes $\mathcal{D}$, and edge text attributes $\mathcal{R}$, a temporal text-attributed graph (TTAG) can be represented as a sequence of interactions $\mathcal{G} = \{(u, v, t)\}$, where $u, v \in \mathcal{V}$ and $t \geq 0$. Each node $u \in \mathcal{V}$ is associated with a node text attribute $d_u \in \mathcal{D}$ and each interaction $(u, v, t) \in \mathcal{G}$ attaches an edge text attribute $r_{u,v,t} \in \mathcal{R}$. We use $\mathcal{H}_u(t) = \{(u, v, \tau) | \tau < t\} \cup \{(v, u, \tau) | \tau < t\}$ to denote the set of historical interactions involving node $u$ before time $t$.

*Definition 2.* **Temporal Text-attributed Graph Modeling.** Given node $u$, time $t$, and all available historical interactions before $t$, $\{(u', v', \tau) | \tau < t\}$, temporal text-attributed graph modeling aims to learn a mapping function $f : (u, t) \mapsto \mathbf{z}_u(t)$, where $\mathbf{z}_u(t) \in \mathbb{R}^d$ denotes the representation of node $u$ at time $t$, and $d$ is the vector dimension.

*Definition 3.* **Language Model *vs.* Large Language Model.** We clearly distinguish between pre-trained language models (LMs) and large language models (LLMs). LMs, *e.g.*, MiniLM, are relatively smaller models designed to embed texts from the text space into the feature space; and LLMs refer to significantly larger models far surpassing the linguistic capabilities of LMs, like DeepSeek-v2/3.

### 2.2 Related Works

We provide two groups of related works: TGNNs and LLM-for-Graph-Text-Augmentation (LLM4GTAug) methods. As illustrated in Fig. 2, we summarize these methods from the perspective of LLM utilization and modality fusion. A discussion of related works is detailed in Sec. E.

**TGNNs.** LLM utilization within TGNNs remains limited. Existing TGNNs [12] typically pre-embed the raw texts as input features and rely on their structural encoders [9] to capture the dynamics of

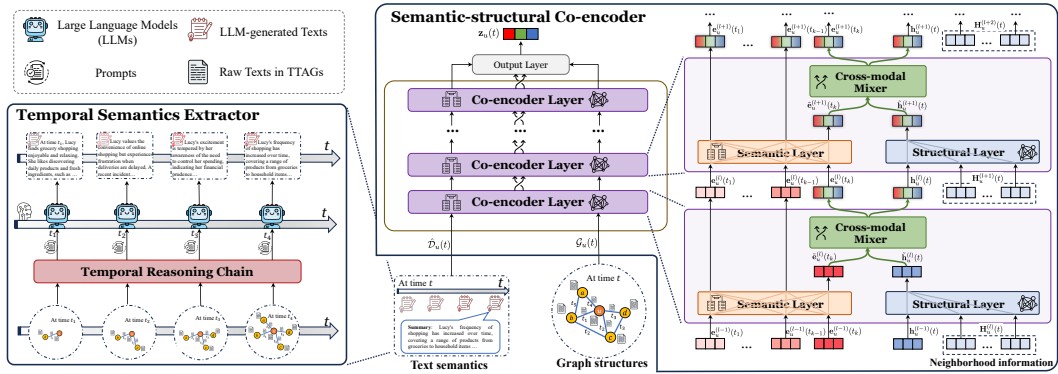

Figure 3: **Architecture of the proposed framework, CROSS.** Our Temporal Semantics Extractor constructs the temporal reasoning chain to empower the LLMs with dynamic reasoning capability to dynamically provide the semantic understanding of nodes' neighborhoods, thereby extracting semantic dynamics in text space. Subsequently, the Semantic-structural Co-encoder iteratively mixes and consolidates both semantic and structural information, ensuring their mutual reinforcement for effectively generating modal-cohesive representations.

graph structures. These methods overlook the temporal evolution of text semantics within TTAGs. Besides, they integrate semantic and structural information only at the input stage, resulting in shallow modality fusion that occurs before encoding.

**LLM4GTAug methods.** Existing LLM4GTAug methods [20, 21] are designed for static graphs and employ LLMs through static, one-off reasoning for each node. This makes them ill-suited for capturing the temporal dynamics of TTAGs. In addition, they mix structural and semantic representations only at the output stage, leading to shallow fusion after encoding.

As summarized in Tab. 1, in this paper, our CROSS aims to (i) enhance LLMs with dynamic reasoning capabilities to capture semantic dynamics; and (ii) facilitate hierarchical modality fusion throughout the entire encoding process. Our proposed method fundamentally opens new avenues for future research in TTAG modeling.

Table 1: **Summary of related works.**

|  | **LLM4GTAug** | **TGNNs** | **CROSS** |
|---|---|---|---|
| Structural Dynamics | ✗ | ✓ | ✓ |
| Semantic Dynamics | ✗ | ✗ | ✓ |
| LLM Utilization | static | no LLM | **dynamic** |
| Modality Fusion | shallow at output (after encoding) | shallow at input (before encoding) | **hierarchical (during encoding)** |

## 3  Methodology

As mentioned before, unifying text semantics and graph structures for TTAG modeling requires considering both semantic dynamics and semantic-structural reinforcement. To this end, as depicted in Fig. 3, the proposed CROSS framework comprises two stages, *i.e.*, the Temporal Semantics Extractor first detects the dynamic text semantics for nodes' evolving neighborhoods; then the Semantic-structural Co-encoder jointly unifies both semantic and structural information to ensure cross-modal reinforcement. We will introduce these two stages in the following subsections.

### 3.1  Temporal Semantics Extractor

In this section, we propose the temporal reasoning chain to advance the LLMs with dynamic reasoning capability to extract semantic dynamics for TTAG modeling. It will be described below.

**Temporal Reasoning Chain.** Existing LLM utilization paradigm within LLM4GTAug methods are ill-suited for semantic extraction in TTAGs, as it fails to effectively capture the dynamics of node's semantic contexts. To address this challenge, we design a novel temporal reasoning chain that empowers the LLMs with the dynamic semantic summarization reasoning capability across temporal dimensions, discerning the evolving linguistic nuances along timestamps. However, performing LLM reasoning at every timestamp remains computationally impractical. To ensure scalability and stability, for each node among TTAGs, we strategically sample reasoning timestamps at equal interaction intervals. This strategy enables CROSS to constrain the number of LLM calls and guarantee that each LLM reasoning step accesses a balanced set of interactions, striking a delicate balance between temporal granularity and cost efficiency.

Formally, for node $u$, we first derive the set of its interaction timestamps $\mathcal{T}_u = \{t_1, t_2, ..., t_n\}$, where $t_1 \leq t_2 \cdots \leq t_n$ and $n$ is the node degree. Then we sample the reasoning timestamps at intervals of $\lceil \frac{n}{m} \rceil$ among $\mathcal{T}_u$ with a predefined maximum reasoning count $m$ as follows:

$$\hat{\mathcal{T}}_u = \left\{ \hat{t}_i \,\middle|\, \hat{t}_i = \begin{cases} t_{i \cdot \lceil \frac{n}{m} \rceil}, & i = 1, \ldots, m-1, \\ t_n, & i = m \end{cases} \in \mathcal{T}_u \right\}. \tag{1}$$

Here, $\hat{\mathcal{T}}_u \subseteq \mathcal{T}_u$ depicts the LLM reasoning timestamps to summarize text semantics around node $u$, and we will also analyze the dataset-specific hyper-parameter $m$ in Sec. I of the Appendix.

**Summarization.** To align with the dynamics of semantic contexts within TTAGs, we facilitate the LLMs to dynamically summarize nodes' neighborhoods across different timestamps. Specifically, we perform multi-turn calling with the LLMs along the previously sampled reasoning timestamps. For node $u$ at reasoning time $\hat{t} \in \hat{\mathcal{T}}_u$, we first obtain its neighborhood by retrieving $u$'s historical interactions $\mathcal{H}_u(\hat{t}) = \{(u, v, \tau) | \tau < \hat{t}\} \cup \{(v, u, \tau) | \tau < \hat{t}\}$, and then we prompt the LLMs to generate neighborhood summaries and semantic details for that time. This can be represented by:

$$\hat{d}_u(\hat{t}) = \text{LLM}\left(d_u, \hat{t}, \{r_*\}_{* \in \mathcal{H}_u(\hat{t})}; \text{PROMPT}\right). \tag{2}$$

$\text{LLM}(\cdot; \text{PROMPT})$ denotes the LLM calling with the prompt, which comprises a fixed template alongside variable pieces, including node $u$'s raw text $d_u$, time $\hat{t}$, and the neighborhood $\{r_*\}_{* \in \mathcal{H}_u(\hat{t})}$. $\hat{d}_u(\hat{t})$ is the LLM-generated text, denoting the semantic summary for $u$'s contexts at time $\hat{t}$. We present a simplified example of PROMPT for clarity, and the complete version is given in Sec. F of the Appendix.

> **A simplified example of PROMPT**
>
> **Goal:** [Request to summarize the current neighborhoods and emphasize the response format.]
> **Descriptions:** [Provide the text attribute of node $d_u$.]
> **Current time:** [Specify the reasoning timestamp $\hat{t}$.]
> **Historical interactions:** [List textualized neighborhoods using recent interactions $\{r_*\}_{* \in \mathcal{H}_u(\hat{t})}$.]

After performing all reasoning, we can obtain a set of LLM-generated textual summaries for each node $u$. We represent them as $\hat{\mathcal{D}}_u = \{\hat{d}_u(\hat{t}) \mid \hat{t} \in \hat{\mathcal{T}}_u\}$. Notably, to incorporate the raw texts in TTAGs, we set $\hat{d}_u(0) = d_u$. These summaries encapsulate the evolving semantics around nodes derived from the dynamically-enhanced LLMs, promoting semantic dynamics for TTAG modeling.

### 3.2 Semantic-structural Co-encoder

As mentioned in the Introduction, existing TGNNs solely account for structural dynamics, overlooking the necessary consideration of semantic dynamics and the reinforcement between these two modalities. To address this issue, our Semantic-structural Co-encoder performs iterative integration of semantics and structures at the layer level. Each layer of the proposed Co-encoder comprises three types of encoding components: (i) *the semantic layer* that encodes the text semantics from LLM-generated summaries to produce semantic representations; (ii) *the structural layer* that encodes the graph structures from neighborhood information to capture structural representations; and (iii) *the cross-modal mixer* that facilitates transformation between these two unimodal representations to cheer their mutual reinforcement. We will discuss them in detail below.

**Semantic Layer.** To better incorporate the semantic modality, we implement our semantic layer using a Transformer encoder [22] due to its widespread usage in textual modeling. From the ablation results in Tab. 3, such a design effectively harnesses the full potential of text semantics for TTAG modeling, demonstrating clear advantages over purely structural encoding mechanisms.

We first prepare the inputs for the semantic layer. For node $u$ at time $t$, the inputs of the semantic layer are the corresponding LLM-generated summaries before $t$ from our Extractor. We represent them as $\hat{\mathcal{D}}_u(t) = \{\hat{d}_u(t_k) \mid t_k < t\}$, where $t_1 \leq \cdots \leq t_k$. We then use a pre-trained LM, such as MiniLM [13], to embed the texts into the $d$-dimensional semantic features, which is depicted as $\hat{\mathbf{s}}_u(t_k) = \text{LM}\left(\hat{d}_u(t_k)\right) \in \mathbb{R}^d$. To capture the temporal differences, we subsequently concatenate time information into these semantic features as follows:

$$\mathbf{x}_u(t_k) = \hat{\mathbf{s}}_u(t_k) \,\|\, \Phi(t - t_k) \in \mathbb{R}^{2d}. \tag{3}$$

Here, $\Phi(\cdot)$ is the time encoding function introduced by [23], which is widely used in recent TGNNs [9]. These enriched features then feed into the $L$-layer Transformer encoder blocks to derive the semantic representations for node $u$ at time $t$. We have:

$$\tilde{\mathbf{e}}_u^{(l)}(t_1), \ldots, \tilde{\mathbf{e}}_u^{(l)}(t_k) = \mathtt{TRM}^{(l)}\left(\mathbf{e}_u^{(l-1)}(t_1), \ldots, \mathbf{e}_u^{(l-1)}(t_k)\right), \tag{4}$$

where $\tilde{\mathbf{e}}_u^{(l)}(t) \in \mathbb{R}^{2d}$ corresponds to the pre-mixed semantic representation at the $l$-th layer and $\mathbf{e}_u^{(0)}(t_1), \ldots, \mathbf{e}_u^{(0)}(t_k) = \mathbf{x}_u(t_1), \ldots, \mathbf{x}_u(t_k)$. As we will mention below, the latest semantic representation of node $u$, $\mathbf{e}_u^{(l-1)}(t_k)$, has integrated information from its structural representation in the previous layer. This allows the $l$-th semantic layer to receive the information from graph structures, achieving deep fusion and promoting their synergistic reinforcement.

**Structural Layer.** Meanwhile, we conduct our structural layer to encode the graph structures around node $u$ at time $t$, $\mathcal{G}_u(t)$, using the structural encoding block in TGNNs. Such a design renders our framework inherently TGNN-agnostic and flexibly integrable with any TGNN backbone.

Formally, the $l$-th structural layer aggregates the neighborhood information of node $u$ from current layer $\mathbf{H}_u^{(l)}(t)$ and its structural representation from the previous layer $\mathbf{h}_u^{(l-1)}(t)$ with a 2-layer Multi-Layer Perceptron (MLP). This can be expressed as:

$$\tilde{\mathbf{h}}_u^{(l)}(t) = \mathtt{MLP}^{(l)}\left(\mathbf{h}_u^{(l-1)}(t) \,\|\, \mathtt{AGG}\left(\mathbf{H}_u^{(l)}(t)\right)\right). \tag{5}$$

Here, $\tilde{\mathbf{h}}_u^{(l)}(t)$ is the pre-mixed structural representation at the $l$-th layer and $\|$ denotes concatenation. $\mathtt{AGG}(\cdot)$ is a pooling function to aggregate the neighborhood information matrix into a $d$-dimensional vector, with specific implementations (*e.g.*, mean, sum) varying across various TGNNs [4, 5, 7].

The neighborhood information $\mathbf{H}_u^{(l)}(t)$ is derived using a temporal attention mechanism [4] via message passing from the $l$-th layer neighborhood. This process assigns attention weights to scale the contribution and importance of neighbors $\mathcal{N}_u(t)$ for node $u$ as follows:

$$\mathbf{H}_u^{(l)}(t) = \mathtt{Softmax}\left(\mathbf{a}_u^{(l)}(t)\right) \cdot \mathbf{V}_u^{(l)}(t), \tag{6}$$

where $\mathbf{a}_u^{(l)}(t) = [a_{uv}^{(l)}(t)]_{v \in \mathcal{N}_u(t)}$ represents the attention weight vector, and the matrix $\mathbf{V}_u^{(l)}(t) = [\mathbf{v}_v^{(l)}(t)]_{v \in \mathcal{N}_u(t)}$ condenses the messages from $u$'s $l$-th layer neighborhood. Each element $a_{uv}^{(l)}(t)$ in $\mathbf{a}_u^{(l)}(t)$ is the attention weight for node $u$ to its neighbor $v \in \mathcal{N}_u(t)$, and each row $\mathbf{v}_v^{(l)}(t)$ from $\mathbf{V}_u^{(l)}(t)$ is the message carried from $u$'s neighboring node $v$. These can be computed by:

$$a_{uv}^{(l)}(t) = \frac{f_{\mathsf{q}}\left(\mathbf{h}_u^{(l-1)}(t)\right) f_{\mathsf{k}}\left(\mathbf{h}_v^{(l-1)}(t)\right)^T}{\sqrt{d^{(k)}}}, \quad (7) \qquad\qquad \mathbf{v}_v^{(l)}(t) = f_{\mathsf{v}}(\mathbf{h}_v^{(l-1)}(t)). \tag{8}$$

In Eqs. 7 & 8, $f_*(\cdot) (* \in \{\mathsf{q}, \mathsf{k}, \mathsf{v}\})$ denote the encoding functions for queries, keys, and values, respectively [22]. These functions may be implemented differently across various TGNNs [4, 5, 7, 9], and we do not discuss their details as they are beyond the scope of our work.

As we explain in the next paragraph, the structural representations from the previous layer, $\mathbf{h}_u^{(l-1)}(t)$, have integrated with the semantic representations. Consequently, the message-passing aggregation in Eq. 5 enables the propagation between both types of information, effectively facilitating semantic-structural reinforcement during encoding.

**Cross-modal Mixer.** Finally, to enable the deep unification between the semantic and structural modalities for TTAG modeling, we introduce a novel and interesting cross-modal mixer that automatically transfers and integrates the bimodal information at a layer-wise granularity.

For node $u$ at the $l$-th layer, the inputs of our cross-modal mixer are the most recent pre-mixed semantic representation $\tilde{\mathbf{e}}_u^{(l)}(t_k)$ and the corresponding pre-mixed structural representation $\tilde{\mathbf{h}}_u^{(l)}(t)$. This means that the remaining historical semantic representations, *i.e.*, $\tilde{\mathbf{e}}_u^{(l)}(t_1), \ldots \tilde{\mathbf{e}}_u^{(l)}(t_{k-1})$, will be not involved in the mixture process. Such a design is driven by: (i) efficiency consideration; (ii) temporal-awareness consideration that the latest semantic representation carries the most contextually relevant information; and (iii) empirical consideration that mixing all semantic representations does not necessarily lead to better performance (See Sec. 4.5). Therefore, we concatenate $\tilde{\mathbf{e}}_u^{(l)}(t_k)$ with

$\tilde{\mathbf{h}}_u^{(l)}(t)$, then pass through our cross-modal mixer, and finally split the fused representation to derive the post-mixed semantic and structural representations. These processes can be formulated as follows:

$$\mathbf{e}_u^{(l)}(t_k); \; \mathbf{h}_u^{(l)}(t) = \texttt{Mixer}^{(l)}\left(\tilde{\mathbf{e}}_u^{(l)}(t_k) \parallel \tilde{\mathbf{h}}_u^{(l)}(t)\right). \tag{9}$$

We implement $\texttt{Mixer}^{(l)}(\cdot)$ using a 2-layer MLP, and it can be alternatively conducted in other fusion components. By iteratively performing such a mixture operation, we can deeply unify both text semantics and graph structures, enabling cohesive representations for TTAG modeling.

### 3.3 Training CROSS

**Cohesive Representations.** For node $u$ at time $t$, its representation is derived from: (i) semantic outputs from the $L$-th semantic layer with mean pooling, $\mathbf{z}_u^{\mathrm{sem}}(t) = \texttt{Mean}\left(\tilde{\mathbf{e}}_u^{(L)}(t_1), \ldots, \tilde{\mathbf{e}}_u^{(L)}(t_k)\right)$; (ii) structural outputs from the $L$-th structural layer, $\mathbf{z}_u^{\mathrm{str}}(t) = \tilde{\mathbf{h}}_u^{(L)}(t)$; and (iii) the unified outputs from the cross-modal mixer, $\mathbf{z}_u^{\mathrm{mix}}(t) = \mathbf{e}_u^{(L)}(t_k) \parallel \mathbf{h}_u^{(L)}(t)$. This can be denoted as:

$$\mathbf{z}_u(t) = \texttt{MLP}_{\mathrm{out}}\left(\mathbf{z}_u^{\mathrm{sem}}(t) \parallel \mathbf{z}_u^{\mathrm{str}}(t) \parallel \mathbf{z}_u^{\mathrm{mix}}(t)\right), \tag{10}$$

where $\mathbf{z}_u(t) \in \mathbb{R}^d$ and $\texttt{MLP}_{\mathrm{out}}(\cdot)$ is a 2-layer MLP that maps the dimension of the input vector to $d$.

**Loss Function.** We adopt the temporal link prediction task [7] as training signals for TTAG modeling. For link $(u, v, t)$, we compute its occurrence probability $\hat{p}_{uv}(t)$ by feeding the concatenated representations of nodes $u$ and $v$, $\mathbf{z}_u(t) \parallel \mathbf{z}_v(t)$, into a 2-layer MLP. Cross-entropy loss is then applied:

$$\mathcal{L} = -\sum_{(u,v,t) \in \mathcal{G}} \left[\log \hat{p}_{uv}(t) + \log\left(1 - \hat{p}_{uv'}(t)\right)\right]. \tag{11}$$

The $v'$ denotes the randomly sampled negative destination node. Additionally, we will also present the theoretical analysis to prove the effectiveness of CROSS in Sec. B.

## 4 Experiments

### 4.1 Experimental Settings

**Datasets.** We implement experiments with five datasets, including four public datasets and one industrial dataset from real-world e-commerce systems. The four public datasets - Enron, GDELT, ICEWS1819, and Googlemap_CT - are recently collected and released by [12]. Besides, the industrial dataset is constructed with three months of transaction data sampled from a private e-commerce trading network in WeChat[2] Mobile Payment.[3] Details of these datasets are summarized in Sec. D.1 due to page limitations. All datasets are chronologically split by 60%, 20%, and 20% for training, validation, and testing, respectively.

**Baselines.** We select eleven existing methods as our baselines, including JODIE [24], DyRep [25], TGAT [4], TGN [5], CAWN [26], PINT [27], TCL [28], GraphMixer [29], DyGFormer [7], LKD4DyTAG [30], and FreeDyG [10]. We also select powerful DeepSeek-v2 [19] and evaluate its zero-shot and one-shot performance as LLM baselines, denoted as $\text{LLM}_{\text{zero/one}}$. Detailed descriptions of all baselines are provided in Sec. D.2. Moreover, we choose three representative TGNN models, *i.e.*, TGAT, TGN, and DyGFormer, as the backbones of CROSS due to their superiority. For simplicity, we employ the well-established MiniLM [13] to embed texts into feature space. Additionally, we adopt DeepSeek-v2 as the default LLM. We also report the results with other LLM backbones in Sec. 4.5. We evaluate the learned representations using two downstream tasks, *i.e.*, temporal link prediction and node classification in an industrial application of financial risk management.

### 4.2 Temporal Link Prediction

We begin our experimental evaluation by comparing the temporal link prediction performance of our model with baselines. We conduct this under two settings: (i) **transductive** setting, which predicts links between nodes that have appeared during training; and (ii) **inductive** setting, where predictions are performed with unseen nodes. Implementation details can be found in Sec. D.3 of the Appendix.

The results are presented in Tab. 2. Clearly, our CROSS framework significantly improves the performance of all three TGNN backbones across all datasets in both transductive and inductive

---

[2]https://pay.weixin.qq.com

[3]The dataset is sampled solely for experimental purposes and does not imply any commercial affiliation. All personally identifiable information (PII) has been removed.

Table 2: **MRR results (%) for temporal link prediction in transductive and inductive settings.** The results for $\textbf{LLM}_{\textbf{zero/one}}$ represent the zero-/one-shot performance of the LLM DeepSeek-v2 [19]. Results highlighted with a blue background indicate the performance and corresponding improvements of our CROSS framework using various TGNN backbones. The best results are highlighted in **bold**.

| | Transductive setting | | | | | Inductive setting | | | | |
|---|---|---|---|---|---|---|---|---|---|---|
| | Enron | GDELT | ICEWS1819 | Googlemap_CT | Industrial | Enron | GDELT | ICEWS1819 | Googlemap_CT | Industrial |
| JODIE | 66.69 ± 2.0 | 48.81 ± 1.1 | 71.47 ± 4.0 | 56.72 ± 0.7 | 49.75 ± 1.2 | 53.41 ± 2.5 | 37.90 ± 2.4 | 57.75 ± 7.1 | 55.21 ± 1.0 | 30.38 ± 0.3 |
| DyRep | 58.85 ± 7.9 | 45.61 ± 2.8 | 63.13 ± 3.6 | 49.04 ± 2.2 | 36.49 ± 0.9 | 42.95 ± 8.3 | 42.23 ± 3.1 | 50.42 ± 2.8 | 47.44 ± 2.5 | 25.47 ± 1.8 |
| TCL | 71.16 ± 0.7 | 59.49 ± 0.5 | 87.58 ± 0.3 | 68.98 ± 0.4 | 50.87 ± 0.5 | 55.28 ± 1.5 | 47.13 ± 1.0 | 77.06 ± 0.2 | 66.26 ± 0.2 | 33.42 ± 0.5 |
| CAWN | 74.56 ± 0.6 | 57.00 ± 0.2 | 82.93 ± 0.1 | 65.34 ± 0.2 | 63.58 ± 0.8 | 61.58 ± 2.0 | 43.56 ± 0.6 | 70.65 ± 0.1 | 62.11 ± 0.2 | 53.42 ± 0.5 |
| PINT | 74.82 ± 2.8 | 52.71 ± 2.5 | 83.81 ± 0.9 | 72.94 ± 0.7 | 53.51 ± 0.6 | 56.38 ± 3.9 | 31.82 ± 4.1 | 63.16 ± 2.8 | 70.02 ± 0.6 | 39.72 ± 0.6 |
| GraphMixer | 62.68 ± 1.3 | 53.33 ± 0.4 | 80.69 ± 0.3 | 53.11 ± 0.2 | 50.50 ± 0.5 | 43.75 ± 1.5 | 41.18 ± 0.3 | 67.09 ± 0.5 | 51.36 ± 0.2 | 34.06 ± 0.7 |
| FreeDyG | 81.52 ± 1.8 | 68.27 ± 0.7 | 86.31 ± 0.6 | 78.82 ± 1.2 | 75.91 ± 0.7 | 70.38 ± 0.1 | 52.71 ± 0.3 | 74.16 ± 0.4 | 66.01 ± 2.8 | 56.48 ± 0.6 |
| LKD4DyTAG | 73.18 ± 0.3 | 57.28 ± 1.9 | 80.62 ± 4.2 | 77.11 ± 0.5 | 77.73 ± 0.7 | 67.45 ± 1.9 | 45.75 ± 2.0 | 73.81 ± 0.3 | 60.73 ± 1.0 | 57.91 ± 1.0 |
| $\text{LLM}_{\text{zero}}$ | 24.18 | 7.99 | 33.68 | 30.30 | 11.27 | 17.73 | 10.08 | 32.26 | 38.21 | 2.62 |
| $\text{LLM}_{\text{one}}$ | 46.27 | 28.91 | 50.82 | 48.79 | 30.29 | 48.14 | 28.91 | 44.69 | 43.83 | 20.28 |
| TGAT | 66.06 ± 0.1 | 56.73 ± .04 | 85.81 ± 0.2 | 63.13 ± 0.5 | 46.74 ± 3.9 | 47.80 ± 0.8 | 42.01 ± 0.5 | 74.10 ± 0.2 | 60.96 ± 0.2 | 30.04 ± 3.0 |
| TGAT+ | 95.58 ± 0.7 | **81.63 ± 1.7** | 93.05 ± 1.6 | 99.91 ± 0.0 | 86.97 ± 2.8 | 81.52 ± 2.0 | 64.56 ± 1.8 | 82.25 ± 2.0 | 91.59 ± 0.0 | 62.22 ± 2.1 |
| *Avg.* ↑ 26.59 | ↑ 29.52 | ↑ **24.90** | ↑ 7.24 | ↑ 36.78 | ↑ 40.23 | ↑ 33.72 | ↑ 22.55 | ↑ 8.15 | ↑ 30.63 | ↑ 32.18 |
| TGN | 73.05 ± 1.7 | 54.28 ± 1.6 | 84.79 ± 0.6 | 71.35 ± 0.5 | 54.46 ± 3.0 | 54.98 ± 2.3 | 37.48 ± 2.8 | 69.69 ± 0.8 | 67.88 ± 0.2 | 38.28 ± 4.1 |
| TGN+ | **95.84 ± 0.4** | 77.95 ± 2.8 | 94.74 ± 5.7 | **99.92 ± 0.0** | 94.26 ± 0.8 | 82.38 ± 1.2 | 56.65 ± 3.8 | 84.01 ± 9.2 | **92.68 ± 0.1** | **83.23 ± 2.6** |
| *Avg.* ↑ 25.45 | ↑ **22.79** | ↑ 23.67 | ↑ 9.95 | ↑ **28.57** | ↑ 39.80 | ↑ 27.40 | ↑ 19.17 | ↑ 14.32 | ↑ **24.80** | ↑ **44.02** |
| DyGFormer | 79.93 ± 0.1 | 61.35 ± 0.3 | 87.51 ± 0.3 | 54.82 ± 2.7 | 74.45 ± 0.7 | 66.86 ± 0.1 | 50.61 ± 0.2 | 78.14 ± 0.3 | 52.98 ± 2.5 | 54.20 ± 0.4 |
| DyGFormer+ | 95.31 ± 2.8 | 81.28 ± 4.4 | **95.77 ± 0.3** | 99.82 ± 0.0 | **94.78 ± 1.4** | **86.01 ± 4.9** | **66.37 ± 4.4** | **87.80 ± 0.6** | 91.74 ± 0.1 | 82.30 ± 1.2 |
| *Avg.* ↑ 22.03 | ↑ 15.38 | ↑ 19.93 | ↑ **8.26** | ↑ 44.99 | ↑ **20.33** | ↑ 19.15 | ↑ **15.76** | ↑ 9.66 | ↑ 38.76 | ↑ 28.10 |

settings. Meanwhile, CROSS achieves SOTA performance with a substantial margin over the best baseline. This observation proves the effectiveness of unifying text semantics and graph structures in TTAG modeling. Although LLM's zero/one-shot performance is suboptimal, CROSS still performs well. This suggests that LLMs may struggle to directly comprehend the dynamics of graph structures in TTAG modeling, but our proposed CROSS framework helps them to resolve this issue effectively. Moreover, our framework tends to result in closer performance across different TGNN backbones. We infer that this is due to the robustness of text semantics, which successfully reduces model reliance on simplistic structural information. Inspired by this, we detail a robustness study in Sec. 4.4.

### 4.3 Industrial Application

We conduct another downstream task using node classification in a real-world industrial application of financial risk management on Industrial, where we predict whether nodes are involved in fraudulent activities. DyGFormer [7] is used as the backbone, and details can be seen in Sec. D.3 of the Appendix. CROSS achieves the best performance as shown in Fig. 4, indicating that the learned representations of CROSS are also effective for node-level tasks. CROSS's success on Industrial demonstrates its *scalability* for large industrial-scale TTAGs. We will also provide scalability clarification in Sec. G.

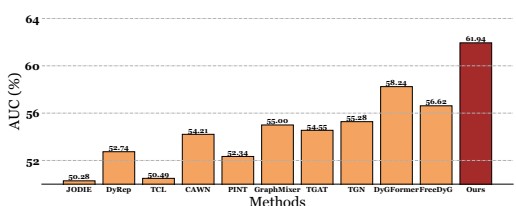

Figure 4: **AUC results (%) for node classification of a real-world industrial application in financial risk management**, which predicts whether nodes are involved in fraudulent activities on Industrial.

### 4.4 Robustness Study

**Robustness for noise.** To empirically validate the assumption of noisy structural information mentioned in Sec. 1, we strategically introduce noise into graph structures with perturbation rates of $p \in \{10\%, 20\%, 30\%, 40\%, 50\%\}$. Implementation details are put in Sec. J due to page limitations. To further investigate the impact of noise during encoding, we randomly select a subset of nodes and visualize the attention weights of their perturbed neighbors using box plots under $p = 50\%$.

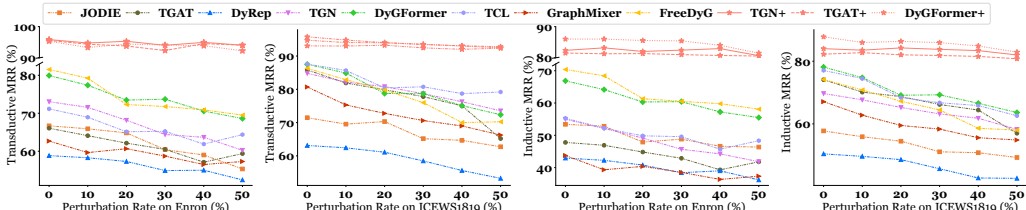

Figure 5: **Robustness study for noise** on Enron and ICEWS1819 with different perturbation rates.

Table 3: **Ablation study for the raw texts and LLM-generated texts. Semantic/Structural Encoding** refer to the encoding mechanisms that independently perform semantic/structural layers in Sec. 3.2; **imprv.** indicates the performance improvements of LLM-generated texts **Text$_{LLM}$** over the raw texts **Text$_{raw}$**. Our proposed co-encoding mechanism unlocks the full potential of LLM-generated texts and achieves the best performance.

| | Datasets | Methods | Semantic Encoding | | | Structural Encoding | | | Semantic-structural Co-encoding | | |
|---|---|---|---|---|---|---|---|---|---|---|---|
| | | | Text$_{raw}$ | Text$_{LLM}$ | imprv. | Text$_{raw}$ | Text$_{LLM}$ | imprv. | Text$_{raw}$ | Text$_{LLM}$ (ours) | imprv. |
| Transductive | Enron | TGAT | 49.73 ± 0.8 | 63.07 ± 0.5 | ↑ 13.34 | 66.06 ± 0.1 | 63.65 ± 1.7 | ↓ 2.41 | 70.27 ± 0.2 | **95.58 ± 0.7** | ↑ 25.31 |
| | | TGN | 49.73 ± 0.8 | 63.07 ± 0.5 | ↑ 13.34 | 73.05 ± 1.7 | 72.36 ± 4.0 | ↓ 0.69 | 74.28 ± 0.9 | **95.84 ± 0.4** | ↑ 21.56 |
| | | DyGFormer | 49.73 ± 0.8 | 63.07 ± 0.5 | ↑ 13.34 | 79.93 ± 0.1 | 80.46 ± 0.5 | ↑ 0.53 | 80.91 ± 0.1 | **95.31 ± 2.8** | ↑ 14.40 |
| | ICEWS1819 | TGAT | 77.45 ± 0.5 | 85.04 ± 1.5 | ↑ 7.59 | 85.81 ± 0.2 | 86.12 ± 0.1 | ↑ 0.31 | 87.33 ± 1.0 | **93.05 ± 1.6** | ↑ 5.72 |
| | | TGN | 77.45 ± 0.5 | 85.04 ± 1.5 | ↑ 7.59 | 84.79 ± 0.6 | 85.69 ± 0.4 | ↑ 0.90 | 85.96 ± 0.8 | **94.74 ± 5.7** | ↑ 8.78 |
| | | DyGFormer | 77.45 ± 0.5 | 85.04 ± 1.5 | ↑ 7.59 | 87.51 ± 0.3 | 88.11 ± 0.5 | ↑ 0.60 | 86.72 ± 0.4 | **95.77 ± 0.3** | ↑ 9.05 |
| Inductive | Enron | TGAT | 31.94 ± 0.7 | 45.24 ± 1.1 | ↑ 13.30 | 47.80 ± 0.8 | 45.01 ± 1.3 | ↓ 2.79 | 53.26 ± 1.0 | **81.52 ± 2.0** | ↑ 28.26 |
| | | TGN | 31.94 ± 0.7 | 45.24 ± 1.1 | ↑ 13.30 | 54.98 ± 2.3 | 53.93 ± 4.0 | ↓ 1.05 | 58.92 ± 1.4 | **82.38 ± 1.2** | ↑ 23.46 |
| | | DyGFormer | 31.94 ± 0.7 | 45.24 ± 1.1 | ↑ 13.30 | 66.86 ± 0.1 | 67.64 ± 1.4 | ↑ 0.78 | 68.27 ± 0.1 | **86.01 ± 4.9** | ↑ 17.74 |
| | ICEWS1819 | TGAT | 60.63 ± 0.8 | 71.45 ± 0.6 | ↑ 10.82 | 74.10 ± 0.2 | 74.12 ± 0.2 | ↑ 0.02 | 75.19 ± 0.2 | **82.25 ± 2.0** | ↑ 7.06 |
| | | TGN | 60.63 ± 0.8 | 71.45 ± 0.6 | ↑ 10.82 | 69.69 ± 0.8 | 70.39 ± 1.2 | ↑ 0.70 | 70.01 ± 0.6 | **84.01 ± 9.2** | ↑ 13.99 |
| | | DyGFormer | 60.63 ± 0.8 | 71.45 ± 0.6 | ↑ 10.82 | 78.14 ± 0.3 | 77.70 ± 0.8 | ↓ 0.44 | 79.79 ± 0.4 | **87.80 ± 0.6** | ↑ 8.01 |

We report the results in Figs. 5 and 6. Our CROSS framework consistently performs best and exhibits remarkable robustness even under high perturbation rates. This may be due to CROSS's ability to effectively harness the valuable temporal semantic information, which aids in mitigating the adverse impact of structural noise attacks. Furthermore, the results of attention weights reveal that the CROSS framework can effectively down-weight the noisy neighbors

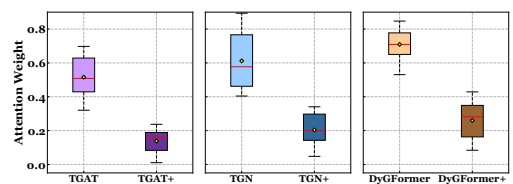

Figure 6: **Attention weights from randomly selected nodes to their perturbed neighbors** on GDELT with the perturbation rate of 50%.

during encoding. This may be the key reason why CROSS could achieve superior performance and exceptional robustness. Building on this observation, we conduct a case study to visualize the attention weights and the learned representations during encoding in Sec. A.

**Robustness for encoding layers.** Next, we study the model robustness to the number of encoding layers. Specifically, we conduct a series of experiments with varying numbers of encoding layers $L = \{1, 2, 3, 4, 5\}$. A larger number of encoding layers facilitates a deeper integration of the cross-modal information. Other details are provided in Sec. J of the Appendix.

The results are depicted in Fig. 7. Our CROSS framework also reveals outstanding robustness to the encoding layers, where the performance improvements over their respective backbones become more pronounced as the number of layers increases. Such robustness likely stems from the invaluable semantic information and the sufficient fusion between semantics and struc-

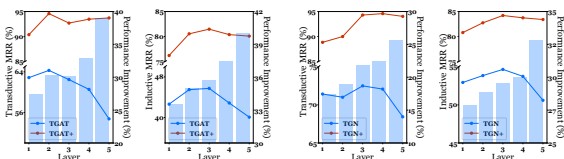

(a) TGAT as the backbone.  (b) TGN as the backbone.

Figure 7: **Robustness study for encoding layers** on Enron.

tures, whereas graph structures alone may carry high-order irrelevant or spurious information. We also find that the CROSS framework always achieves peak performance with a larger $L$. This can be attributed to the deeper information exchange of our cross-modal mixer, which fully amplifies the mutual reinforcement between semantics and structures.

### 4.5 Ablation Study

We conduct three groups of ablation experiments, including model components, textual inputs and their encoding strategies, as well as the choice of LLMs.

**Ablation for model components.** We start the ablation study by evaluating the contributions of the key components of our model. Detailed information for each variant is provided in Sec. K of Appendix. The results are detailed in Fig. 8. We can see that incorporating all components results in the best performance, while the removal of any single component leads to a performance drop. This highlights the effectiveness of each

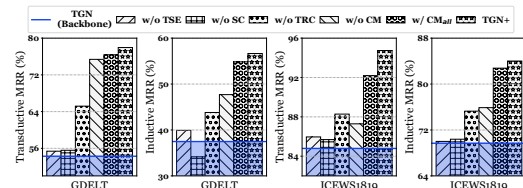

Figure 8: **Ablation study for model components** using TGN model as the backbone.

component in CROSS. Notably, mixing all semantic representations does not improve model performance. This may be attributed to the unexpected inclusion of overly outdated semantic information when passing through our cross-modal mixer, thus hindering the quality of final representations.

**Ablation for raw texts and LLM-generated texts.** We then conduct an ablation study to compare the **raw texts** (*i.e.*, the original text attributes in TTAGs) with the **LLM-generated texts** (*i.e.*, neighborhood summaries produced by the LLMs) using various encoding mechanisms. We present the results in Tab. 3 and other details can be seen in Sec. K. Firstly, our framework outperforms all other variants, reaffirming its effectiveness. Besides, the results under semantic encoding prove the validity and expressiveness of the pure LLM-generated texts. Interestingly, the performance of LLM-generated texts yields only marginal improvements or even slight degradation when performing structural encoding. We infer that this occurs because the structural encoding introduces excessively irrelevant information from high-order relations, whereas the other two encodings directly capitalize on the node's own LLM-generated texts, thus integrating more focused and relevant information. This strongly demonstrates the necessity and superiority of the design of our co-encoding mechanism.

**Ablation for different LLMs.** We extend our ablation study with various LLMs, including DeepSeek-v2 [19] (default), GPT-4o [31], Llama3-8b [32], Vicuna-7b [33], and Mistral-7b [34]. Details and results are put in Sec. K and Fig. 9, respectively. We find that DeepSeek-v2 performs best and all variants outperform their corresponding TGNN backbones, reaffirming the effectiveness of the proposed CROSS framework.

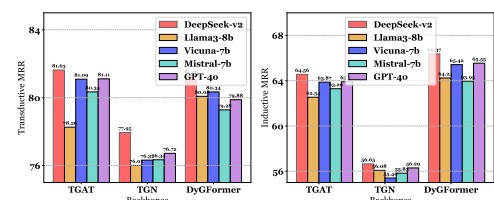

Figure 9: **Ablation study for different LLMs** on GDELT.

## 5   Conclusion and Future Work

In this paper, we focus on the under-explored problem of TTAG modeling and propose CROSS, which extends existing TGNNs to effectively unify text semantics and graph structures with LLMs. By introducing the Temporal Semantics Extractor, we can enhance the LLMs to dynamically extract the text semantics within nodes' neighborhoods. The Semantic-structural Co-encoder then integrates semantic and structural information, enabling bidirectional reinforcement between both modalities. As for future work, we will consider more complex designs of our cross-modal mixer for achieving better representation fusion, such as using the time decay mechanism.

## Acknowledgments

This work is funded in part by the NSFC under grant No. 62372013 and No. 62206056, the NSF through grant IIS-2106972, the CIPSC-SMP-Zhipu Large Model Cross-Disciplinary Fund, and also supported by Tencent Wechat Group. The authors would like to express their gratitude to the reviewers for their feedback, which has improved the clarity and contribution of the paper. The first author, Dr. Zhang, also wants to thank Yifeng Wang for his efforts and support in this work.

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

Figure 10: **Case study for text semantics and graph structures.** Thicker edges denote higher attention weights during encoding. **CROSS exhibits exceptional prediction performance and robustness** by successfully unifying text semantics and graph structures within TTAGs, which enables the model to adaptively adjust the attention weights to concentrate on more relevant neighbors, thereby achieving improved prediction performance.

## A  Case Study

In this section, we conduct a case study to qualitatively investigate the effectiveness of CROSS.

**Case study for text semantics and graph structures.** As illustrated in Fig. 10, we select a representative node from Googlemap_CT and visualize its raw texts, the LLM-generated texts from Temporal Semantics Extractor, the attention weights assigned among neighborhoods by both the TGAT backbone and CROSS during encoding, as well as the prediction probability. The value of prediction probability refers to the predicted probability for the corresponding positive edges as defined in Eq. 11, where higher values indicate better performance. We find that CROSS demonstrates a remarkable capability to unify semantics and structures, which allows the model to adaptively adjust attention weights to concentrate on more relevant neighbors during encoding, thereby achieving improved prediction performance. For instance, CROSS effectively detects the semantic shift in preferences for the target node "*Ryan Blanck*" across temporal dimensions, which is from "*restaurant*" to "*shopping*" ($t_2 \rightarrow t_3$). Subsequently, it automatically reduces the attention weights of neighbors assigned to restaurant nodes (*e.g.*, "*Frank Pepe Pizzeria Napoletana*") while increasing the weight for store-related neighbors (*e.g.*, "*Stafford Thrift Shop*"). This highlights the **complementarity** of semantics and structures, enabling CROSS to prioritize preferred neighbors with semantic contexts.

**Case study for the learned representations.**
We further conduct a case study for the learned representations directly. Specifically, we extract semantic and structural representations of a selected node among GDELT, visualize their distributions using Kernel Density Estimation (KDE), and compare the differences between CROSS and the backbone. As shown in Tab. 11, CROSS redistributes the feature space and produces more cohesive representations between semantics and structures. This can be attributed to our co-encoding architecture, which facilitates synergistic reinforcement between

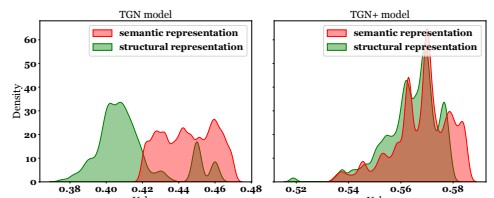

Figure 11: **Case study for the learned representations.** CROSS effectively synthesizes more cohesive, unified representations between semantics and structures.

modalities. Additionally, we find that semantic and structural representations from backbones have different distributions. This directly proves the **complementarity** of the two modalities, as each captures distinct information that contributes to the final representations. As a result, CROSS effectively integrates them and leads to unified representations that incorporate the most salient features of both.

## B    Theoretical Analysis

CROSS is designed from the extraction and unification of semantics and structures within TTAGs. In this section, we present a theoretical analysis for this idea to prove the effectiveness of CROSS, showing that unifying text semantics and graph structures yields more expressive information than existing TGNNs that rely solely on graph structures. This holds under two key conditions:

- *Integrity*: Text semantics can accurately reflect the graph structures among the neighborhoods of the target node, as they are derived from the LLM-generated texts enriched by the dynamically-informed LLMs' parameterized knowledge.
- *Complementarity*: Text semantics complement the graph structures among the neighborhoods of the target node, providing additional contextual cues that help distinguish subtle relationships.

**Theorem 1.** *Consider the conditions as follows:*

*1) Integrity: $Z_T$ serves as a reliable proxy for $Z_G$, thus we have*

$$H\left(Z_G|Z_T\right) = \epsilon, \quad \epsilon > 0. \tag{12}$$

*2) Complementarity: $Z_T$ encapsulates orthogonal information that is not captured by $Z_G$, and that is*

$$H\left(y|Z_G, Z_T\right) = H\left(y|Z_G\right) - \epsilon', \quad \epsilon' > \epsilon. \tag{13}$$

*Under these conditions, it follows that:*

$$H\left(y|Z_G, Z_T\right) < H\left(y|Z_G\right), \tag{14}$$

*where $Z_G$ denotes the information derived from graph structures, $Z_T$ represents the information captured from text semantics, $y$ is the target for prediction, and $H\left(\cdot|\cdot\right)$ depicts the condition entropy.*

We also provide an alternative theoretical analysis to prove the model effectiveness, showing that unifying text semantics and graph structures instead of solely considering graph structures is bounded.

**Theorem 2.** *Given the target $y$, there exists a constant $\beta \in (0, 1]$ such that:*

$$I(Z_G, Z_T; y) \geq I(Z_G; y) + \beta \min\left\{H(y|Z_G), \; H(Z_T|Z_G)\right\}, \tag{15}$$

*where $Z_G$ denotes the information derived from graph structures, $Z_T$ represents the information captured from text semantics, $I(\cdot|\cdot)$ denotes mutual information, and $H(\cdot|\cdot)$ depicts the condition entropy.*

*Proof of Theorem 1.* We aim to prove that the conditional entropy of $y$ unifying both graph structures $Z_G$ and text semantics $Z_T$, *i.e.*, $H\left(y|Z_G, Z_T\right)$, is strictly less than the conditional entropy of $y$ solely based on $Z_G$, *i.e.*, $H\left(y|Z_G\right)$.

We begin with:

$$H(y|Z_G, Z_T). \tag{16}$$

Next, we decompose this using the properties of entropy into two phases:

$$H(y|Z_G, Z_T) = H(y|Z_G, Z_M, Z_T) + I(y; Z_M|Z_G, Z_T), \tag{17}$$

where $Z_M$ denotes the information arising from the mixed representations of text semantics and graph structures.

We then apply the following upper bound on conditional mutual information as follows:

$$\begin{aligned} I(y; Z_M|Z_G, Z_T) &= H(Z_M|Z_G, Z_T) - H(Z_M|y, Z_G, Z_T) \\ &\leq H(Z_M|Z_G, Z_T). \end{aligned} \tag{18}$$

Here, the first equality follows from the definition of mutual information, and the inequality holds due to the nonnegativity of conditional entropy.

By substituting Eq. 18 into Eq. 17, we obtain:

$$H(y|Z_G, Z_T) \leq H(y|Z_G, Z_M, Z_T) + H(Z_M|Z_G, Z_T). \tag{19}$$

Since conditional entropy increases when conditioning on fewer variables, it follows that:

$$H(y|Z_G, Z_M, Z_T) + H(Z_M|Z_G, Z_T) \leq H(y|Z_G, Z_T) + H(Z_G|Z_T). \tag{20}$$

By applying the "Integrity" and "Complementarity" conditions, we arrive at:

$$H(y|Z_G, Z_M) + H(Z_M|Z_T) \leq H(y|Z_G) - \epsilon' + \epsilon. \tag{21}$$

Finally, since $\epsilon' > \epsilon$, we conclude:

$$H(y|Z_G) - \epsilon' + \epsilon < H(y|Z_G). \tag{22}$$

Consequently, we have proven that:

$$H(y|Z_G, Z_T) < H(y|Z_G). \tag{23}$$

This completes the proof. □

*Proof of Theorem 2.* We aim to prove that the mutual information of $y$ unifying both graph structures $Z_G$ and text semantics $Z_T$, *i.e.*, $I(Z_G, Z_T; y)$, is lower bounded by the mutual information of $y$ solely based on $Z_G$, i.e., $I(Z_G; y)$.

We begin with:

$$I(Z_G, Z_T; y). \tag{24}$$

Using the chain rule of mutual information, $I(Z_G, Z_T; y)$ can be decomposed as:

$$I(Z_G, Z_T; y) = I(Z_G; y) + I(Z_T; y \mid Z_G), \tag{25}$$

where $I(Z_G; y)$ quantifies the contribution of graph structures $Z_G$ about $y$, and $I(Z_T; y \mid Z_G)$ reflects the additional information provided by text semantics $Z_T$ conditioned on $Z_G$.

Then, we compute the lower bound for the term $I(Z_T; y \mid Z_G)$. Based on the fundamental properties of entropy, we can obtain:

$$\begin{aligned} I(Z_T; y \mid Z_G) &= H(y \mid Z_G) - H(y \mid Z_T, Z_G) \\ &= H(Z_T \mid Z_G) - H(Z_T \mid y, Z_G). \end{aligned} \tag{26}$$

Here, the conditional mutual information $I(Z_T; y \mid Z_G)$ could be as high as the minimum of the conditional entropies $H(y \mid Z_G)$ and $H(Z_T \mid Z_G)$. Considering any unavoidable loss in information during the unification process, there exists a constant $\beta \in (0, 1]$, such that:

$$I(Z_T; y \mid Z_G) \geq \beta \min\{H(y \mid Z_G), \ H(Z_T \mid Z_G)\}. \tag{27}$$

Substituting the lower bound from Eq. 27 into Eq. 25, we obtain the final bound:

$$\begin{aligned} I(Z; y) &= I(Z_G; y) + I(Z_T; y \mid Z_G) \\ &\geq I(Z_G; y) + \beta \min\{H(y|Z_G), H(Z_T|Z_G)\}. \end{aligned} \tag{28}$$

This completes the proof. □

## C  Notations and Algorithms

We provide the important notations used in this paper and their corresponding descriptions as shown in Tab. 4. Additionally, for clarity, we present the pseudo-codes of CROSS in Algorithm 1.

## D  Details of Experimental Setting

### D.1  Datasets

In this paper, we select four public datasets [12] from different domains and one real-world industrial dataset. We present their detailed descriptions below, and their statistics are summarized in Tab. 5.

---
**Algorithm 1:** Training CROSS (one epoch).
---
**Input:** A node set $\mathcal{V}$; A TTAG $\mathcal{G} = \{(u, v, t)\}$ with node text attributes $\mathcal{D}$ and edge text
        attributes $\mathcal{R}$; The maximum reasoning count $m$; The number of encoding layer $L$.
---
**1** Initialize all model parameters and prepare the LLMs;
   `// Temporal Semantics Extractor`
**2** **foreach** $u \in \mathcal{V}$ **do**
**3**     Derive reasoning timestamps $\hat{\mathcal{T}}_u$ with $m$ by Eq. 1;
**4**     Summarize $u$'s textualized neighborhood at $\hat{t} \in \hat{\mathcal{T}}_u$ with LLM by Eq. 2;
**5** **end**
   `// Semantic-structural Co-encoder`
**6** **foreach** *batch* $(u, v, t) \subseteq \mathcal{G}$ **do**
**7**     **foreach** $l = 1, 2, ..., L$ **do**
**8**         Retrieve text semantics from generated summaries and graph structures from
           neighborhoods for nodes $u/v$;
**9**         Compute pre-mixed semantic representations $\tilde{\mathbf{e}}^{(l)}_{u/v}(t_k)$ with semantic layer by Eqs. 3-4;
**10**        Compute pre-mixed structural representations $\tilde{\mathbf{h}}^{(l)}_{u/v}(t)$ with structural layer by Eqs. 5-8;
**11**        Mix and propagate cross-modal representations by Eq. 9;
**12**     **end**
**13**     Derive the final representations $\mathbf{z}_{u/v}(t)$ by Eq. 10;
**14**     Compute loss $\mathcal{L}$ by Eq. 11 and backward;
**15** **end**
---

Table 4: **Important notations and descriptions.**

| Notations | Descriptions |
|:---:|:---:|
| $d_u$ | Raw text attribute of node $u$ |
| $r_{u,v,t}$ | Raw text attribute of edge $(u, v, t)$ |
| $\hat{d}_u(\hat{t})$ | LLM-generated text summary for $u$'s neighborhood at reasoning time $\hat{t}$ |
| $\tilde{\mathbf{e}}^{(l)}_u(t)$ | Pre-mixed semantic representation for node $u$ at time $t$ in the $l$-th layer |
| $\mathbf{e}^{(l)}_u(t)$ | Post-mixed semantic representation for node $u$ at time $t$ in the $l$-th layer |
| $\tilde{\mathbf{h}}^{(l)}_u(t)$ | Pre-mixed structural representation for node $u$ at time $t$ in the $l$-th layer |
| $\mathbf{h}^{(l)}_u(t)$ | Post-mixed structural representation for node $u$ at time $t$ in the $l$-th layer |
| $\mathbf{z}_u(t)$ | Final representation for node $u$ at time $t$ |

- **Enron**[4] originates from a collection of email exchanges among employees of the ENRON energy corporation spanning three years (1999–2002). In this dataset, nodes represent employees, and edges correspond to emails exchanged between them. Each node has text attributes that are derived from the employee's department and role if such information is available. Each edge attaches text attributes consisting of the raw content of the emails. Edges are sequentially ordered based on the e-mail sending timestamps.

- **GDELT**[5] originates from the Global Database of Events, Language, and Tone, a project aimed at cataloging political behaviors across nations worldwide. In this dataset, nodes represent political entities, such as "Egypt" or "Kim Jong Un". The textual attributes of nodes are directly taken from the names of these entities. Edges capture the relationships between entities (e.g., "Make Empathetic Comment" or "Provide Aid"), with the textual attributes of edges being derived from the descriptions of these relationships. Edges are sequentially ordered based on the event-occurring timestamps.

- **ICEWS1819**[6] is sourced from the Integrated Crisis Early Warning System project, which serves as a larger temporal knowledge graph for tracking political events compared to Enron.

---

[4]`https://www.cs.cmu.edu/~enron`
[5]`https://www.gdeltproject.org`
[6]`https://dataverse.harvard.edu/dataverse/icews`

This dataset is built using events occurring between January 1, 2018, and December 31, 2019. The textual attributes of nodes include the name, sector, and nationality of each political entity, while the textual attributes of edges represent descriptions of the political relationships. All edges are sequentially ordered based on the event-occurring timestamps.

- **Googlemap_CT**[7] is sourced from the Google Local Data project, which compiles review data from Google Maps along with user and business information in the United States up to September 2021. This dataset specifically focuses on business entities located in Connecticut. Nodes represent users and businesses, while edges correspond to user reviews of businesses. Textual attributes are assigned exclusively to business nodes, encompassing the business name, address, category, and self-introduction. All edges are sequentially ordered based on the review timestamps.

- **Industrial** is sourced from real-world e-commerce transaction records sampled from a mobile payment company, spanning March to June 2024. Nodes in this dataset represent users or merchants while edges denote their transaction records. Each node is enriched with text attributes, such as the user/merchant name and affiliation. Text attributes of each edge include textual details such as price, transaction type, and user review. Besides, all edges are sequentially ordered based on the transaction timestamps, and each node is assigned a **label** indicating whether it is fraudulent.

Table 5: **Detailed statistics of datasets.**

| Datasets | # Nodes | # Links | # Times | Duration | Domains | Time Granularity |
|---|---|---|---|---|---|---|
| Enron | 42,711 | 797,907 | 1,006 | 3 years | E-mail | one day |
| GDELT | 6,786 | 1,339,245 | 2,591 | 2 years | knowledge graph | 15 minutes |
| ICEWS1819 | 31,769 | 1,100,071 | 730 | 2 years | knowledge graph | 24 hours |
| Googlemap_CT | 111,168 | 1,380,623 | 55,521 | – | Recommendation | Unix Time |
| Industrial | 1,112,094 | 3,196,008 | 90 | 3 months | E-commerce | one day |

## D.2 Baselines

We evaluate the performance and discuss the capabilities of eleven existing TGNN methods. The details of these methods are as follows:

- **JODIE** [24] is designed to manage temporal graphs in bipartite user-item settings. It employs two Recurrent Neural Networks (RNNs), one for updating the user states and another for the item states. To prevent the issue of outdated node representations, a projection layer is added to track the evolution of the embeddings over time.

- **DyRep** [25] incorporates neighborhood information by utilizing a temporal attention-based aggregation mechanism. This approach helps capture the evolving structural features of nodes' local environments in the temporal graph, allowing for more accurate dynamic representations.

- **TGAT** [4] leverages a temporal attention model to aggregate data from temporal-topological neighbors, facilitating the creation of temporal node embeddings. It also introduces a trainable time encoding function that ensures each temporal step is distinctly represented, a concept widely adopted in later TGN architectures.

- **TGN** [5] builds upon earlier methodologies by introducing a memory system that stores a state vector for each node. This memory is refreshed whenever a node participates in an interaction. The model also features modules for processing messages, updating memory states, and embedding temporal features, which collectively enable the generation of dynamic node representations.

- **CAWN** [26] creates node embeddings using temporal walks. It generates multiple anonymous random walks starting from a target node and encodes them using a Recurrent Neural Network. These encoded walks are then combined to form the final temporal representation, which is particularly effective for predicting temporal links.

---

[7]https://datarepo.eng.ucsd.edu/mcauley_group/gdrive/googlelocal

- **TCL** [28] uses a breadth-first search to form temporal dependency sub-graphs, extracting sequences of interactions for analysis. A Transformer encoder is applied to integrate temporal and structural information, enabling central node representation learning. Additionally, TCL incorporates a cross-attention mechanism within the Transformer to capture interdependencies between interacting node pairs.
- **PINT** [27] applies injective message passing with a temporal focus and incorporates relative positional encoding to improve the model's capability in capturing dynamic patterns within neighborhoods.
- **GraphMixer** [29] employs a link encoder inspired by the MLP-Mixer framework to create temporal embeddings for nodes. Its design includes a fixed time encoding scheme, which demonstrates superior performance compared to traditional learnable approaches. The model also utilizes a node encoder with mean-pooling to aggregate link-based information.
- **DyGFormer** [7] relies on information from 1-hop neighbors to learn temporal graph representations. A Transformer encoder with a patching method is used to capture long-range dependencies across nodes. To preserve correlations between source and target nodes, DyGFormer integrates a Neighbor Co-occurrence Feature.
- **LKD4DyTAG** [30] conducts a preliminary exploration of dynamic text-attributed graphs. It leverages LLMs as text embedders and introduces an auxiliary knowledge distillation loss to enhance model performance.
- **FreeDyG** [10] delves the temporal graph modeling into the frequency domain and proposes a node interaction frequency encoding module that both effectively models the proportion of the re-occurred neighbors and the frequency of corresponding interactions of the node pair.

In addition to the above existing deep learning-based TGNNs, we also explore the performance of the LLMs for TTAG modeling. We employ the widely adopted and well-performing LLM, DeepSeek-v2[8] [19], an open-source project released by DeepSeek, Inc.[9]. DeepSeek-v2 is a strong, economical, and efficient mixture-of-experts language model. For comparison, we test its zero-shot and one-shot performance for temporal link prediction, which is denoted as $\text{LLM}_{\text{zero}}$ and $\text{LLM}_{\text{one}}$, respectively. Similar to our temporal reasoning chain in Sec. 3.1, we design a task-specific prompt to call with LLMs. Specifically, given the historical interactions of two nodes, we prompt DeepSeek-v2 to directly predict whether these two nodes will interact at a specific future timestamp.

### D.3 Implementation Details

**Tasks and Metrics.** We follow [6] and conduct **temporal link prediction** under two settings: (i) transductive setting, which predicts links between nodes that have appeared during training; and (ii) inductive setting, where predictions are performed with unseen nodes. During training, we sample an equal number of negative destination nodes as described in Eq. 11. Inspired by [8], we employ Mean Reciprocal Rank (MRR) as the evaluation metric with 100 negative links per positive link during evaluation. We also report the AP and AUC results in Tabs. 7 & 8. Additionally, we further conduct the **node classification** task in a practical industrial application for financial risk management using the Industrial dataset from the e-commerce domain. The objective of this task is to predict whether a node is involved in fraudulent activity. Specifically, we follow [6] and pass the learned representations through a two-layer MLP to get the probabilities of fraudulent activity for each node. We adopt the Area Under the Receiver Operating Characteristic Curve (AUC) as the evaluation metric for this task.

**Model Configurations.** For the **training and evaluation**, we follow [12] and train all models for 50 epochs and adopt the early stopping strategy under the patience of 5 with an evaluation interval of 5. The learning rate and the batch size across all models and datasets are set to 0.0001 and 256, respectively. We repeat the experiments for 3 runs with seeds ranging from 0 to 2 to ensure evaluation reliability and report the averaged performance with the corresponding standard deviations. All training is performed on a single server with 72 cores, 128GB memory, and four Nvidia Tesla V100 GPUs. As for the **hyper-parameters**, the representation dimensions across all models and datasets are consistently set to 384, and the introduced hyper-parameter, maximum reasoning count $m$, is set to 8 for all datasets by default. Other hyper-parameters among baselines follow the critical

---

[8] https://github.com/deepseek-ai/DeepSeek-V2
[9] https://deepseek.com

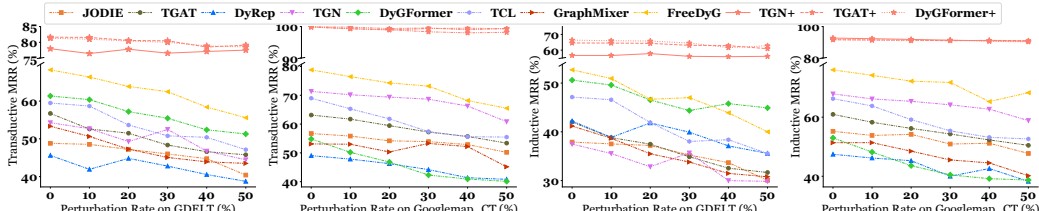

Figure 12: **Robustness study for noise** on GDELT and Googlemap_CT with different perturbation rates. (Supplementary results for Fig. 5.)

Table 6: **Ablation study for the raw texts and LLM-generated texts. Semantic/Structural Encoding** refer to the encoding mechanisms that independently perform semantic/structural layers in Sec. 3.2; ***imprv.*** indicates the performance improvements of LLM-generated texts **Text$_{LLM}$** over the raw texts **Text$_{raw}$**. (Supplementary results for Tab. 3.)

|  | Datasets | Methods | Semantic Encoding | | | Structural Encoding | | | Structural-Structural Co-encoding | | |
|---|---|---|---|---|---|---|---|---|---|---|---|
|  |  |  | Text$_{raw}$ | Text$_{LLM}$ | *imprv.* | Text$_{raw}$ | Text$_{LLM}$ | *imprv.* | Text$_{raw}$ | Text$_{LLM}$(ours) | *imprv.* |
| Transductive | GDELT | TGAT | 49.64 ± 0.9 | 52.02 ± 8.8 | ↑ 2.38 | 56.73 ± .04 | 57.01 ± 0.3 | ↑ 0.28 | 58.29 ± 0.2 | **81.63 ± 1.7** | ↑ 23.34 |
|  |  | TGN | 49.64 ± 0.9 | 52.02 ± 8.8 | ↑ 2.38 | 54.28 ± 1.6 | 55.58 ± 1.0 | ↑ 1.30 | 56.27 ± 1.2 | **77.95 ± 2.8** | ↑ 21.68 |
|  |  | DyGFormer | 49.64 ± 0.9 | 52.02 ± 8.8 | ↑ 2.38 | 61.35 ± 0.3 | 62.54 ± 0.1 | ↑ 1.19 | 62.73 ± 0.5 | **81.28 ± 4.4** | ↑ 18.55 |
|  | Googlemap_CT | TGAT | 47.03 ± 0.2 | 98.38 ± 0.4 | ↑ 51.35 | 63.13 ± 0.5 | 68.69 ± 0.1 | ↑ 5.56 | 65.46 ± 0.2 | **99.92 ± 0.0** | ↑ 34.46 |
|  |  | TGN | 47.03 ± 0.2 | 98.38 ± 0.4 | ↑ 51.35 | 71.35 ± 0.5 | 81.60 ± 1.0 | ↑ 10.25 | 72.64 ± 0.8 | **99.92 ± 0.0** | ↑ 27.28 |
|  |  | DyGFormer | 47.03 ± 0.2 | 98.38 ± 0.4 | ↑ 51.35 | 54.82 ± 2.7 | 61.76 ± 0.1 | ↑ 6.94 | 57.02 ± 0.4 | **99.82 ± 0.0** | ↑ 42.80 |
|  | Industrial | TGAT | 24.47 ± 0.5 | 73.15 ± 1.8 | ↑ 48.68 | 46.74 ± 3.9 | 53.33 ± 2.7 | ↑ 6.59 | 47.62 ± 2.0 | **86.97 ± 2.8** | ↑ 39.35 |
|  |  | TGN | 24.47 ± 0.5 | 73.15 ± 1.8 | ↑ 48.68 | 54.46 ± 3.0 | 53.20 ± 1.0 | ↓ 1.26 | 55.45 ± 0.5 | **94.26 ± 0.8** | ↑ 38.81 |
|  |  | DyGFormer | 24.47 ± 0.5 | 73.15 ± 1.8 | ↑ 48.68 | 74.45 ± 0.7 | 74.05 ± 0.4 | ↓ 0.40 | 75.23 ± 0.1 | **94.78 ± 1.4** | ↑ 19.55 |
| Inductive | GDELT | TGAT | 27.98 ± 0.7 | 37.72 ± 9.5 | ↑ 9.74 | 42.01 ± 0.3 | 45.79 ± 0.4 | ↑ 3.78 | 41.02 ± 0.3 | **64.56 ± 1.8** | ↑ 23.54 |
|  |  | TGN | 27.98 ± 0.7 | 37.72 ± 9.5 | ↑ 9.74 | 37.48 ± 2.8 | 34.61 ± 3.1 | ↓ 2.87 | 38.81 ± 1.0 | **56.65 ± 3.8** | ↑ 17.84 |
|  |  | DyGFormer | 27.98 ± 0.7 | 37.72 ± 9.5 | ↑ 9.74 | 50.61 ± 0.2 | 52.33 ± .04 | ↑ 1.72 | 52.19 ± 0.3 | **66.37 ± 4.4** | ↑ 14.18 |
|  | Googlemap_CT | TGAT | 44.43 ± 0.2 | 89.53 ± 0.7 | ↑ 45.10 | 60.96 ± 0.2 | 66.98 ± 0.2 | ↑ 6.02 | 62.17 ± 0.2 | **91.59 ± 0.0** | ↑ 29.42 |
|  |  | TGN | 44.43 ± 0.2 | 89.53 ± 0.7 | ↑ 45.10 | 67.88 ± 0.2 | 78.08 ± 1.0 | ↑ 10.20 | 68.17 ± 0.2 | **92.68 ± 0.1** | ↑ 24.51 |
|  |  | DyGFormer | 44.43 ± 0.2 | 89.53 ± 0.7 | ↑ 45.10 | 52.98 ± 2.5 | 59.67 ± 0.1 | ↑ 6.69 | 53.81 ± 0.6 | **91.74 ± 0.1** | ↑ 37.93 |
|  | Industrial | TGAT | 20.71 ± 0.5 | 46.34 ± 2.3 | ↑ 25.63 | 30.04 ± 3.0 | 33.86 ± 1.2 | ↑ 3.82 | 34.19 ± 1.2 | **62.22 ± 2.1** | ↑ 28.03 |
|  |  | TGN | 20.71 ± 0.5 | 46.34 ± 2.3 | ↑ 25.63 | 38.28 ± 4.1 | 34.41 ± 1.4 | ↓ 3.87 | 40.17 ± 2.0 | **83.23 ± 2.6** | ↑ 43.06 |
|  |  | DyGFormer | 20.71 ± 0.5 | 46.34 ± 2.3 | ↑ 25.63 | 54.20 ± 0.4 | 54.18 ± 0.2 | ↓ 0.02 | 55.37 ± 2.7 | **82.30 ± 1.2** | ↑ 26.93 |

hyper-parameters in the widely-used library DyGLib [10] [7], which has performed an exhaustive grid search to identify the optimal hyper-parameters across different models. For the **details of LLM calls**, in addition to the ablation study on different LLMs of Sec. 4.5, we adopt DeepSeek-v2 as the default LLM. All LLM calls on DeepSeek-v2 are performed in a Language Model as a Service (LMaaS)-compatible manner via its official Application Programming Interface (API)[11].

# E   Related Work

**Temporal Graph Neural Networks (TGNNs).** Temporal graph neural networks (TGNNs) [35, 36, 37, 38, 39, 40, 41, 42, 43, 44, 45, 46] are designed to generate node representations in temporal graphs, where they typically develop various structural encoding mechanisms to summarize the dynamic graph structures among neighborhoods of the target node [47, 48, 49, 50, 51, 52, 53, 54, 55, 56]. Based on how these mechanisms operate, existing TGNNs can be broadly categorized into two types: message-encoding TGNNs (ME-TGNNs) and walk-encoding TGNNs (WE-TGNNs). ME-TGNNs [57, 58, 59, 60, 61, 62, 63, 64] capture changing graph structures via message passing mechanisms, where node representation is refined by aggregating messages from neighbors through various aggregation functions [9]. In contrast, WE-TGNNs [26, 27] incorporate temporal structural information into node representations in a different way. They typically sample multiple temporal walks originating from the target node and encode these walks based on node occurrence information. Despite their success, above existing TGNNs focus solely on biased encoding mechanisms that prioritize topological dynamics, neglecting the rich text semantics present in temporal text-attributed graphs (TTAGs) [12].

For completeness, we note the existence of a very recent work [30] that conducts *a preliminary exploration of TTAG modeling*, where LLMs are used to embed texts into features, and a distillation

---

[10]https://github.com/yule-BUAA/DyGLib

[11]https://api-docs.deepseek.com

Table 7: **AP results (%) for temporal link prediction in transductive and inductive settings.** Results highlighted with a ~blue background~ indicate the performance and corresponding improvements of our CROSS framework using various TGNN backbones. The best results are highlighted in **bold**.

| | Transductive setting | | | | | Inductive setting | | | | |
|---|---|---|---|---|---|---|---|---|---|---|
| | Enron | GDELT | ICEWS1819 | Googlemap_CT | Industrial | Enron | GDELT | ICEWS1819 | Googlemap_CT | Industrial |
| JODIE | 97.26 ± 0.2 | 94.87 ± 0.2 | 97.62 ± 0.6 | 84.28 ± 0.1 | 93.70 ± 0.2 | 93.91 ± 0.4 | 91.47 ± 0.2 | 94.40 ± 1.7 | 82.33 ± 0.2 | 82.57 ± 0.5 |
| DyRep | 95.99 ± 1.3 | 93.76 ± 0.5 | 96.04 ± 0.4 | 77.21 ± 1.2 | 87.64 ± 0.2 | 90.99 ± 2.7 | 91.00 ± 0.7 | 91.76 ± 0.5 | 74.44 ± 1.6 | 77.23 ± 0.4 |
| TCL | 97.43 ± 0.1 | 96.16 ± 0.0 | 99.23 ± 0.0 | 89.80 ± 0.2 | 94.33 ± 0.2 | 93.49 ± 0.4 | 92.80 ± 0.1 | 98.17 ± 0.0 | 88.09 ± 0.0 | 82.98 ± 0.3 |
| CAWN | 97.74 ± 0.0 | 95.50 ± 0.1 | 98.90 ± 0.0 | 88.41 ± 0.1 | 96.37 ± 0.1 | 94.48 ± 0.2 | 91.19 ± 0.2 | 97.31 ± 0.0 | 86.29 ± 0.0 | 89.14 ± 0.1 |
| PINT | 98.28 ± 0.1 | 96.26 ± 0.2 | 97.33 ± 0.1 | 90.20 ± 0.1 | 96.00 ± 0.2 | 95.82 ± 0.4 | 92.28 ± 0.4 | 98.11 ± 0.1 | 87.25 ± 0.1 | 90.91 ± 0.2 |
| GraphMixer | 96.27 ± 0.2 | 94.96 ± 0.1 | 98.60 ± 0.0 | 80.12 ± 0.3 | 94.18 ± 0.1 | 90.40 ± 0.6 | 90.79 ± 0.1 | 96.60 ± 0.1 | 77.43 ± 0.1 | 82.66 ± 0.2 |
| FreeDyG | 97.26 ± 0.1 | 95.02 ± 0.1 | 99.08 ± 0.1 | 84.20 ± 0.4 | 96.22 ± 0.3 | 92.46 ± 0.8 | 91.87 ± 0.2 | 97.29 ± 0.3 | 79.22 ± 0.1 | 84.44 ± 0.3 |
| LKD4DyTAG | 98.42 ± 0.1 | 96.26 ± 0.3 | 99.17 ± 0.2 | 84.88 ± 0.1 | 97.20 ± 0.1 | 93.02 ± 0.2 | 91.19 ± 0.4 | 98.21 ± 0.0 | 80.63 ± 0.2 | 85.88 ± 0.5 |
| TGAT | 96.94 ± 0.1 | 95.70 ± 0.1 | 99.10 ± 0.0 | 87.11 ± 0.3 | 93.60 ± 0.6 | 91.98 ± 0.2 | 91.57 ± 0.1 | 97.81 ± 0.0 | 85.40 ± 0.2 | 81.25 ± 0.9 |
| TGAT+ | 99.67 ± 0.1 | 98.11 ± 0.2 | 99.64 ± 0.1 | 99.97 ± 0.0 | 97.51 ± 0.2 | 97.44 ± 0.2 | 94.38 ± 0.3 | 98.63 ± 0.2 | 97.23 ± 0.1 | 93.41 ± 0.5 |
| *Avg.* ↑ 5.55 | ↑ 2.73 | ↑ 2.41 | ↑ 0.54 | ↑ 12.86 | ↑ 3.91 | ↑ 5.46 | ↑ 2.81 | ↑ 0.82 | ↑ 11.83 | ↑ 12.16 |
| TGN | 97.98 ± 0.2 | 95.35 ± 0.3 | 99.05 ± 0.1 | 91.52 ± 0.1 | 95.00 ± 0.6 | 94.58 ± 0.5 | 90.23 ± 0.7 | 97.48 ± 0.1 | 90.00 ± 0.0 | 85.48 ± 1.5 |
| TGN+ | **99.71 ± 0.1** | 98.07 ± 0.3 | 99.73 ± 0.3 | **99.98 ± 0.0** | 97.90 ± 0.3 | 97.60 ± 0.3 | 93.36 ± 0.7 | 98.74 ± 0.7 | **97.45 ± 0.1** | 94.69 ± 0.1 |
| *Avg.* ↑ 4.06 | ↑ **1.73** | ↑ 2.72 | ↑ 0.68 | ↑ **8.46** | ↑ 2.90 | ↑ 3.02 | ↑ 3.13 | ↑ 1.26 | ↑ **7.45** | ↑ 9.21 |
| DyGFormer | 97.90 ± 0.2 | 96.43 ± 0.0 | 99.17 ± 0.0 | 81.16 ± 2.1 | 97.56 ± 0.1 | 94.99 ± 0.3 | 93.45 ± 0.1 | 98.13 ± 0.0 | 78.74 ± 2.4 | 89.81 ± 0.1 |
| DyGFormer+ | 99.63 ± 0.3 | **98.50 ± 0.3** | **99.78 ± 0.0** | 99.97 ± 0.0 | **98.79 ± 0.1** | **98.25 ± 0.6** | **95.83 ± 0.5** | **99.09 ± 0.0** | 97.25 ± 0.1 | **95.90 ± 0.1** |
| *Avg.* ↑ 5.57 | ↑ 1.73 | ↑ **2.07** | ↑ **0.61** | ↑ **18.81** | ↑ 1.23 | ↑ **3.26** | ↑ **2.38** | ↑ **0.96** | ↑ 18.51 | ↑ **6.09** |

Table 8: **AUC results (%) for temporal link prediction in transductive and inductive settings.** Results highlighted with a ~blue background~ indicate the performance and corresponding improvements of our CROSS framework using various TGNN backbones. The best results are highlighted in **bold**.

| | Transductive setting | | | | | Inductive setting | | | | |
|---|---|---|---|---|---|---|---|---|---|---|
| | Enron | GDELT | ICEWS1819 | Googlemap_CT | Industrial | Enron | GDELT | ICEWS1819 | Googlemap_CT | Industrial |
| JODIE | 97.50 ± 0.2 | 95.55 ± 0.2 | 97.56 ± 0.6 | 85.25 ± 0.3 | 94.34 ± 0.2 | 93.93 ± 0.3 | 91.41 ± 0.2 | 93.68 ± 2.0 | 82.83 ± 0.4 | 80.55 ± 0.7 |
| DyRep | 96.56 ± 0.9 | 94.63 ± 0.2 | 95.83 ± 0.3 | 78.02 ± 0.5 | 87.97 ± 0.3 | 91.60 ± 2.1 | 90.71 ± 0.4 | 90.60 ± 0.5 | 74.60 ± 1.0 | 74.25 ± 0.7 |
| TCL | 97.52 ± 0.1 | 96.32 ± 0.0 | 99.18 ± 0.01 | 89.75 ± 0.2 | 94.97 ± 0.2 | 93.20 ± 0.4 | 92.69 ± 0.1 | 98.04 ± 0.02 | 87.95 ± 0.0 | 80.58 ± 0.5 |
| CAWN | 97.76 ± 0.0 | 95.62 ± 0.0 | 98.86 ± 0.01 | 88.51 ± 0.1 | 96.57 ± 0.1 | 94.07 ± 0.1 | 90.92 ± 0.1 | 97.15 ± 0.04 | 86.39 ± 0.1 | 87.17 ± 0.1 |
| PINT | 98.00 ± 0.4 | 96.27 ± 0.2 | 99.01 ± 0.1 | 89.36 ± 0.1 | 98.11 ± 0.3 | 95.35 ± 0.7 | 92.52 ± 0.1 | 98.27 ± 0.5 | 88.24 ± 0.2 | 87.25 ± 0.3 |
| GraphMixer | 96.42 ± 0.4 | 95.26 ± 0.0 | 97.27 ± 0.1 | 80.08 ± 0.3 | 94.81 ± 0.1 | 89.99 ± 0.7 | 90.69 ± 0.1 | 95.27 ± 0.1 | 76.57 ± 0.2 | 80.01 ± 0.4 |
| FreeDyG | 97.72 ± 0.1 | 96.32 ± 0.2 | 99.08 ± 0.01 | 84.37 ± 0.2 | 95.29 ± 0.1 | 95.27 ± 0.9 | 92.23 ± 0.1 | 97.92 ± 0.1 | 80.58 ± 0.1 | 84.29 ± 0.2 |
| LKD4DyTAG | 97.01 ± 0.1 | 97.22 ± 0.1 | 98.00 ± 0.1 | 83.29 ± 0.1 | 96.44 ± 0.0 | 92.82 ± 0.4 | 93.73 ± 0.1 | 96.62 ± 0.1 | 78.76 ± 0.2 | 82.98 ± 0.1 |
| TGAT | 97.17 ± 0.1 | 95.93 ± 0.1 | 99.05 ± 0.0 | 87.17 ± 0.4 | 94.43 ± 0.4 | 92.09 ± 0.3 | 91.74 ± 0.1 | 97.67 ± 0.1 | 85.33 ± 0.2 | 78.88 ± 0.6 |
| TGAT+ | 99.69 ± 0.04 | 98.17 ± 0.2 | 99.62 ± 0.1 | 99.97 ± 0.0 | 97.04 ± 0.2 | 97.53 ± 0.1 | 94.28 ± 0.3 | 98.54 ± 0.3 | 97.20 ± 0.1 | 92.15 ± 0.4 |
| *Avg.* ↑ 5.47 | ↑ 2.52 | ↑ 2.24 | ↑ 0.57 | ↑ 12.80 | ↑ 2.61 | ↑ 5.44 | ↑ 2.54 | ↑ 0.87 | ↑ 11.87 | ↑ 13.27 |
| TGN | 98.15 ± 0.2 | 95.67 ± 0.3 | 99.02 ± 0.1 | 91.96 ± 0.1 | 95.49 ± 0.4 | 94.86 ± 0.5 | 90.53 ± 0.5 | 97.47 ± 0.0 | 90.50 ± 0.0 | 83.51 ± 1.6 |
| TGN+ | **99.70 ± 0.1** | 98.14 ± 0.3 | 99.73 ± 0.3 | 99.97 ± 0.0 | 97.69 ± 0.7 | 97.48 ± 0.4 | 93.69 ± 0.7 | 98.72 ± 0.7 | **97.42 ± 0.0** | 93.17 ± 0.2 |
| *Avg.* ↑ 3.86 | ↑ **1.55** | ↑ 2.47 | ↑ 0.71 | ↑ 8.01 | ↑ 2.20 | ↑ 2.62 | ↑ 3.16 | ↑ 1.25 | ↑ **6.92** | ↑ 9.66 |
| DyGFormer | 97.78 ± 0.4 | 96.52 ± 0.0 | 99.08 ± 0.02 | 80.96 ± 2.2 | 97.61 ± 0.1 | 94.24 ± 0.6 | 93.12 ± 0.1 | 97.93 ± 0.0 | 77.99 ± 2.7 | 87.89 ± 0.2 |
| DyGFormer+ | 99.60 ± 0.3 | **98.53 ± 0.3** | **99.75 ± 0.03** | 99.97 ± 0.0 | **98.24 ± 0.1** | **98.03 ± 0.7** | **95.57 ± 0.5** | **98.99 ± 0.1** | 97.24 ± 0.1 | **94.70 ± 0.1** |
| *Avg.* ↑ 5.75 | ↑ 1.82 | ↑ **2.01** | ↑ **0.67** | ↑ **19.01** | ↑ 0.63 | ↑ **3.79** | ↑ 2.45 | ↑ **1.06** | ↑ 19.25 | ↑ 6.81 |

loss is applied to transfer knowledge from LLMs to a TGNN model. However, this method still suffers from the limitations of neglecting both semantic dynamics and semantic-structural reinforcement, as it continues to rely on the TGNN encoding mechanism during inference (empirically validated in Tab. 2). Instead, our proposed CROSS tackles these issues by unifying text semantics and graph structures, which effectively generates cohesive representations that are both context- and structure-aware.

**Text-attributed Graphs (TAGs).** Text-attributed graphs (TAGs) have been widely adopted in numerous real-world applications [16, 65]. To enable representation learning in such graphs, existing methods often combine graph learning approaches with language modeling techniques. Early works focus on integrating pre-trained language models (LMs) with graph neural networks (GNNs). Some of them [66, 67] conduct cascaded architecture. They first use LMs to independently embed texts as node features, which are then fed into GNNs for representations. Unlike these pipelines, other methods [68, 69, 70, 71, 72, 73] adopt a hybrid GNN-LM architecture to jointly detect both semantic and structural information. Unfortunately, all these methods overlook the potential temporal information inherent in graphs, limiting their applicability to TTAG modeling. Moreover, they cannot be directly applied to TTAGs, as TTAGs and TAGs differ technically. For instance, TTAGs involve a sequence of timestamped interactions with both node and edge attributes, while TAGs typically rely on adjacency matrices with only node attributes.

**Large Language Models (LLMs) for Graph Text Augmentation (LLM4GTAug).** In recent years, the rising prominence of large language models (LLMs) [18, 74, 75], such as DeepSeek-v2/3 [19], has underscored their exceptional potential to revolutionize graph modeling [76, 77, 78]. They typically harness the prompt response of LLMs as the external knowledge to fortify overall model performance, either for text augmentation [20, 79, 80] or for structure refinement [21, 81]. However, all the above methods focus on static text-attributed graphs. LLMs used in these methods are typically limited by static corpora input for reasoning, making them ill-suited to capture the temporal dynamics of TTAGs. In this paper, we propose to enhance LLMs with dynamic semantic summarization reasoning capability, effectively detecting semantic dynamics for TTAG modeling.

## F   Prompt Template

In Sec. 3.1, we provide a simplified example of the prompts used to construct the temporal reasoning chain within our Temporal Semantics Extractor. Here, we present a complete example of the prompts applied to the Googlemap_CT dataset as depicted in Fig. 13. Only minor keyword adjustments are required for the prompts when applied to other datasets. Specifically, for the Enron dataset, the terms "*item*" and "*review*" are replaced with "*user*" and "*email*". Similarly, for the GDELT and ICEWS1819 datasets, these terms are substituted with "*entity*" and "*relation*". For the industrial datasets, "*item*" and "*review*" are replaced with "*user*" and "*transaction*".

> **# Goal #:** Summarize the historical reviews of user 'Maureen Sobel' at the current timestamp and provide your semantic understanding for them.
> **# Descriptions #:** xxx
> **# Current timestamp #:** 27639
> **# Recent reviews of user 'Maureen Sobel' #:**
>    0. timestamp: 27630 | item: 'Name: xxx. Description: xxx' | review: 'Had my hair cut and permed by a very experienced and…'
>    1. timestamp: 27246 | item: 'Name: xxx. Description: xxx' | review: 'Always clean and monitored. Have a good selection of machines …'
>    2. timestamp: 26880 | item: 'Name: xxx. Description: xxx' | review: 'Basic auto parts store with lackadaisical stereotype staff who …'
>    3. …
>       …
> Provide the summary STRICTLY in this form: Summary: xxx.

Figure 13: **An example of the prompt** used to query the LLMs on Googlemap_CT.

## G   Scalability Clarification of LLM Usage

In this section, we provide a scalability clarification of LLM usage within the CROSS framework, including presenting cost statistics and outlining the strategies employed to improve efficiency.

**Cost statistics of LLM usage.** DeepSeek-v2 [19] is renowned for its exceptional performance and cost-efficiency, which provides an excellent balance between quality and affordability. Therefore, we select DeepSeek-v2 as the default LLM in our CROSS framework. Here, we present the cost statistics for calling DeepSeek-v2 API across the five datasets in Tab. 9 for reference. Additionally, we want to emphasize that the DeepSeek-v2 API imposes no theoretical rate limits. During practical implementation, we leverage multithreading techniques to conduct multiple network requests simultaneously under 80 concurrent processes, enhancing program parallelism and optimizing response time consumption. As for the money cost, DeepSeek-v2 prices at $0.00027 per 1,000 input tokens and $0.0011 per 1,000 output tokens at the time of our experiments.

Table 9: **Cost statistics of LLM usage.**

| Datasets | # Input Tokens | # Output Tokens | # Time Consumption (s) | # Money Cost ($) |
|---|---|---|---|---|
| Enron | 20,640,423 | 4,471,765 | 4,528 | 10.49 |
| GDELT | 4,194,490 | 1,049,842 | 4,145 | 2.29 |
| ICEWS1819 | 21,844,799 | 5,410,145 | 6,517 | 11.85 |
| Googlemap_CT | 144,712,376 | 21,214,481 | 21,581 | 62.41 |
| Industrial | 296,273,271 | 30,294,172 | 41,729 | 113.3 |

**Strategies to improve the efficiency of LLM usage.** It is important to highlight that we have implemented several strategies to enhance the efficiency of LLM usage as follows: (i) *The frequency of LLM calls for each node is carefully constrained.* We introduce a maximum reasoning count $m$ to

constrain the number of reasoning steps for each node as described in Eq. 1. This design reduces the computational complexity of LLM calls from $\mathcal{O}(|\mathcal{E}|)$ to $\mathcal{O}(|\mathcal{V}|)$, where $|\mathcal{E}|$ and $|\mathcal{V}|$ represent the total number of edges and nodes respectively. (ii) *Smaller LLMs can serve as cost-effective alternatives.* Smaller LLMs within the CROSS framework still demonstrate notable performance as shown in Sec. 4.5, providing a more cost-effective alternative to larger LLMs like DeepSeek-v2 or GPT-4o. (iii) *The LLM-generated texts are consistently reused to avoid repeated querying.* Our method requires only a single query to the LLMs, with the summaries being stored for subsequent use. These LLM-generated texts can be reused for other tasks or integrated into other methods. We will make the LLM-generated texts publicly available if this paper can be accepted.

Based on the clarifications above, we argue that **the proposed CROSS framework is applicable and effective for large industrial-scale TTAGs**. This is because CROSS's success on our Industrial dataset from real-world e-commerce systems has demonstrated its practical scalability, and the scale of this dataset surpasses that of many common domains [82]. We will also provide the complexity analysis for CROSS's components in Sec. L.4.

## H Efficiency Study

In this section, we conduct an efficiency study by comparing model performance and training time per epoch. The results are presented in Fig. 14, where the top-left indicates better performance with higher efficiency. From the results, we observe that **CROSS strikes a better trade-off between effectiveness and efficiency**, achieving the best performance while maintaining a moderate, acceptable computational cost. These results further highlight the scalability and practicality of our proposed framework, making it well-suited for large-scale and industrial applications.

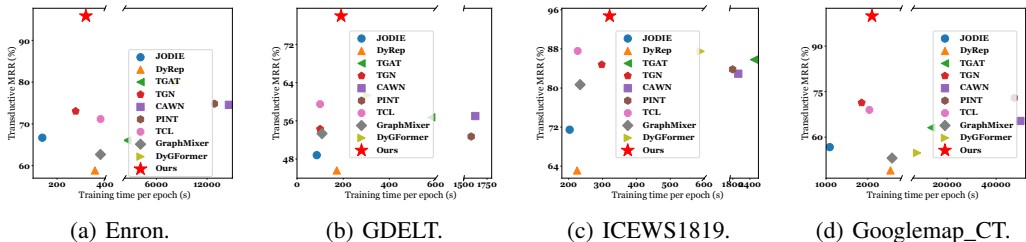

(a) Enron.  (b) GDELT.  (c) ICEWS1819.  (d) Googlemap_CT.

Figure 14: **Comparison between model performance and training time per epoch.** CROSS exhibits a better trade-off between effectiveness and efficiency, achieving the best performance while maintaining a moderate, acceptable computational cost.

## I Parameter Study

To balance granularity and efficiency, we set a maximum reasoning count ($m$ in Eq. 1) to constrain the number of LLM reasoning steps. Now we study how this hyper-parameter impacts performance and plot the results with varying $m$ in Fig. 15. Reasoning too infrequently (small $m$) may make the LLMs cannot effectively comprehend the semantic dynamics around nodes and thus result in degraded performance, while reasoning too frequently

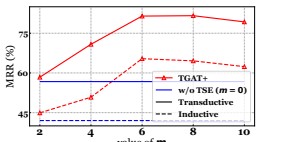  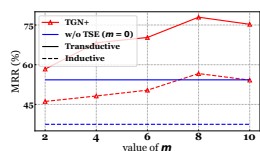

(a) TGAT as the backbone.  (b) TGN as the backbone.

Figure 15: **Parameter study with different maximum reasoning count** $m$ on GDELT. Results of w/o TSE are also presented for reference.

(large $m$) may cause the LLMs to fail to capture long-term semantic shifts. It is also worth noting that different models exhibit varying sensitivities, and $m = 8$ seems to be a generally sweet choice.

## J Details for Robustness Study

**Robustness study for noise.** As stated in Sec. 1, the learned representations from existing TGNNs may rely solely on noisy structural information. To empirically validate this assumption, we strategically introduce noise into graph structures surrounding a target node by replacing its neighbors

with randomly sampled nodes at perturbation rates of $p \in \{10\%, 20\%, 30\%, 40\%, 50\%\}$. To further investigate the impact of noise and assess the model ability to handle such conditions, we randomly select a subset of nodes and visualize the attention weights of their perturbed neighbors during encoding using box plots under $p = 50\%$.

**Robustness study for encoding layers.** Next, we study the model robustness to the number of encoding layers. Specifically, we conduct a series of experiments with varying numbers of encoding layers $L = \{1, 2, 3, 4, 5\}$. A larger number of encoding layers facilitates a deeper integration of the cross-modal information. We employ TGAT and TGN as the backbones. Unlike previous experiments, we set the neighbor sampling size in each layer to 5 in this group of experiments due to computational constraints. Other hyper-parameters remain set to their default values detailed in Sec. D.3.

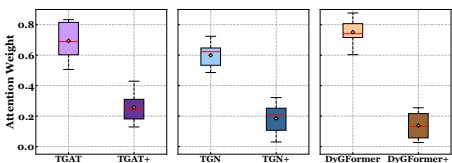

Figure 16: **Attention weights from randomly selected nodes** to their perturbed neighbors on Enron with the perturbation rate of 50%. (Supplementary results for Fig. 6.)

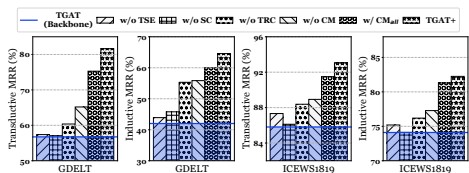

Figure 18: **Ablation study for model components** using TGAT model as the backbone. Results of the backbone are also included for reference. (Supplementary results for Fig. 8.)

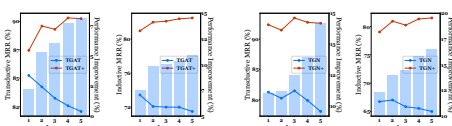

(a) TGAT as backbone.    (b) TGN as backbone.

Figure 17: **Robustness study for encoding layers** on ICEWS1819. (Supplementary results for Fig. 7.)

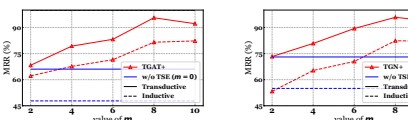

(a) TGAT as backbone.    (b) TGN as backbone.

Figure 19: **Parameter study with different maximum reasoning count** $m$ on Enron. Results of w/o TSE are shown for reference. (Supplementary results for Fig. 15.)

## K    Details for Ablation Study

**Ablation study for model components.** We start the ablation study by evaluating the contributions of the key components of our model, including the Temporal Semantics Extractor (TSE) in Sec. 3.1, the Semantic-structural Co-encoder (SC) in Sec. 3.2, and the Cross-modal Mixer (CM) in Eq. 9. We remove each component individually, resulting in three variants: **w/o TSE**, **w/o SC**, and **w/o CM**. Moreover, we disrupt the Temporal Reasoning Chain constructed by the TSE component via scrambling the chronological order in $\hat{\mathcal{T}}_u$ among Eq. 1, which results in a variant named **w/o TRC**. Additionally, to validate the rationale behind the design of our Cross-modal Mixer, we present results where all semantic representations are indiscriminately mixed. This variant is referred to as **w/ CM$_{all}$**.

**Ablation study for raw texts and LLM-generated texts.** We then conduct an ablation study to compare the **raw texts** in TTAGs (*i.e.*, the original node or edge text attributes) with the **LLM-generated texts** (*i.e.*, neighborhood summaries produced by LLM-based Temporal Semantics Extractor). Experiments are performed using three encoding mechanisms, including the **semantic encoding** that performs semantic layers for semantic representations described in Sec. 3.2, the **structural encoding** that conducts semantic layers for structural representations described in Sec. 3.2, and the **semantic-structural co-encoding** that is used to generate final representations in CROSS. These combinations lead to six variants.

**Ablation study for different LLMs.** We further extend our ablation study with different LLMs. In addition to the default LLM DeepSeek-v2 within the CROSS framework, we conduct experiments using various LLMs with the GDELT dataset, including GPT-4o [31], Llama3-8b [32], Vicuna-7b [33], and Mistral-7b [34]. Specifically, **GPT-4o** is a proprietary large language model developed

by OpenAI[12], built upon the GPT-4 architecture and optimized for multimodal comprehension and generation. We also invoke it via its official API[13]. **Llama3-8b** is an open-source auto-regressive language model with an improved transformer structure. Its tuned versions use supervised fine-tuning (SFT) and reinforcement learning with human feedback (RLHF) to align with human preferences for helpfulness and safety. **Vicuna-7b** is built on Llama 2 and fine-tuned using supervised instruction. Its training data comes from user-shared conversations found online. **Mistral-7b** uses grouped-query attention (GQA) for faster processing and sliding window attention (SWA) to handle long sequences more efficiently, reducing the cost of inference.

From the results in Fig. 9, we can find that DeepSeek-v2 performs best, demonstrating its effectiveness. Additionally, it is worth noting that all variants significantly outperform their corresponding backbones and the performance differences across variants are relatively minimal. This demonstrates the robustness of the CROSS framework.

## L Discussions

### L.1 Empirical Analysis for Semantic Dynamics

In this subsection, we provide an empirical analysis for the semantic dynamics in TTAGs. To achieve this, we conduct an empirical experiments with three additional variants, including:

- **w/ FTI** (Fixed Time Interval): Instead of the timestamp sampling using fixed interaction intervals in Eq. 1, for each node, we compute its interaction time span and uniformly sample timestamps to enable the LLM to summarize at fixed time intervals.
- **w/ RN** (Recent Neighbors): For each node at each timestamp, we directly instruct the LLM to summarize the most recent 20 neighbors.
- **w/o TE** (Time Encoding): We directly remove the time encoding mentioned in Eq. 3.

The results are presented in Tab. 10. From the results, we find that the w/o TE exhibits only a marginal performance drop. This indicates that time encoding alone captures limited temporal dynamics. Instead, our prompting paradigm guides LLMs to capture the semantic evolution over time, effectively facilitating text-driven dynamics for TTAG modeling. Furthermore, either summarizing at fixed time intervals or recent interactions yields suboptimal results. This strongly validates the importance of semantic dynamics.

Table 10: MRR results (%) for temporal link prediction on Enron with TGAT and TGN backbones.

| | Transductive | | Inductive | |
|---|---|---|---|---|
| | TGAT | TGN | TGAT | TGN |
| w/ FTI | 82.47 | 85.20 | 70.21 | 71.77 |
| w/ RN | 78.46 | 83.38 | 65.60 | 68.87 |
| w/o TE | 92.87 | 93.62 | 81.40 | 80.33 |
| CROSS | **95.58** | **95.84** | **81.52** | **82.38** |

### L.2 Extension to Edge Classification

CROSS can be easily extended to edge classification. For completeness, we evaluate two edge classification settings: (i) Supervised setting where models are trained directly using edge labels, and (ii) zero-shot setting where models are first trained on temporal link prediction, and a 2-layer MLP is subsequently fine-tuned on top of the frozen encoder for edge classification. Both implementation and model configurations follow DTGB [12], with DyGFormer adopted as the backbone. From the results, we find that CROSS still achieves the best performance in both supervised and zero-shot edge classification, proving its strong generalization and transferability across tasks. Additionally, all models exhibit a performance drop in the zero-shot setting. This highlights the presence of a transfer learning challenge in TTAG modeling, which warrants future research.

---

[12]https://openai.com
[13]https://openai.com/api

Table 11: Precision, Recall, and F1 (%) results for supervised and zero-shot edge classification on Enron. We omit some baselines, as they have shown inferior performance [12].

| | Supervised Edge Classification | | | | | Zero-shot Edge Classification | | | | |
| | JODIE | DyRep | GraphMixer | DyGFormer | CROSS | JODIE | DyRep | GraphMixer | DyGFormer | CROSS |
|---|---|---|---|---|---|---|---|---|---|---|
| Precision | 65.68 | 66.25 | 63.13 | 66.01 | **78.62** | 47.52 | 50.28 | 44.28 | 52.82 | **65.28** |
| Recall | 64.72 | 63.90 | 57.35 | 58.06 | **72.74** | 40.25 | 46.91 | 42.82 | 47.36 | **60.22** |
| F1 | 64.78 | 64.32 | 55.07 | 56.04 | **70.91** | 42.88 | 44.58 | 43.66 | 45.20 | **58.29** |

### L.3 When Handling Conflicting Modalities

CROSS is developed from the unification of text semantics and graph structures with the assumption of complementarity between both information, which is discussed in Sec. 1 and validated in Sec. A. While building upon this, we argue that CROSS remains effective even in the presence of potential cross-modal conflicts. This claim is supported by:

**Quantitative results.** Noise can reduce alignment and amplify conflicts between semantics and structures. Our robustness study for noise in Sec. 4.4 simulates these inconsistencies, and the results show that CROSS consistently performs best and exhibits strong robustness. This validates CROSS's ability to effectively handle such cross-modal conflicts.

**Qualitative results.** Similar to our case study in Sec. A, we also visualize the learned representations of a representative node on GDELT as shown in Fig.20. The completely disjoint representations from backbones confirm the presence of cross-modal conflicts, and CROSS effectively narrows the feature space to produce relatively more unified representations. Such capabilities to handle conflicts can be attributed to our co-encoding mechanism, which adaptively facilitates synergistic alignment between modalities.

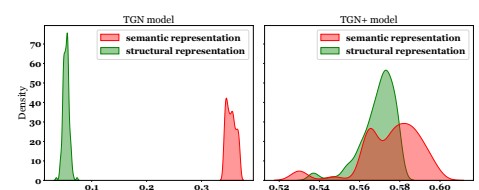

Figure 20: **Case study for the learned representations when handling conflicting modalities**.

### L.4 Complexity Analysis

Now, we provide a theoretical complexity analysis of the two main components of CROSS. Let $|\mathcal{V}|$ represent the number of nodes, $D$ denote the average degree of the node, $L$ indicate the number of encoding layers, and $m$ depict the maximum reasoning count. To simplify the calculations, we assume that the input, hidden, and output dimensions are uniformly set to $d$. In the Temporal Semantics Extractor, we construct a temporal reasoning chain with a maximum reasoning count of $m$ for each node, resulting in a computational complexity of $\mathcal{O}(m|\mathcal{V}|)$. For the Semantic-structural Co-encoder, we utilize various existing TGNN encoding blocks as the structural layers. Therefore, our complexity analysis focuses on the additional cost introduced by the semantic layers. As mentioned in Sec. 3.2, these layers employ a series of standard Transformer layers for each node, thus leading to a total complexity of $\mathcal{O}\left(L|\mathcal{V}|D^2 d + L|\mathcal{V}|d\right)$.

### L.5 Limitation

One potential limitation of CROSS is that we only focus on 1-hop historical interactions of a specific node as input to the LLMs within our Temporal Semantics Extractor. While effective in many cases, this approach may be suboptimal for scenarios where high-order temporal semantics are critical. However, directly incorporating multi-hop textual neighborhoods into LLM could substantially escalate computational costs and dilute the model ability to capture relevant semantic information. Future work could explore innovative methods to efficiently and effectively capture nodes' high-order semantics, unlocking further potential for improved TTAG modeling.

Another issue could be that LLM-generated texts do not always perform optimally under different encoding mechanisms. Although LLMs are renowned for their powerful text generation and understanding capabilities, it is crucial to explore effective encoding mechanisms that can fully maximize the potential of LLM-generated texts, such as the proposed Semantic-structural Co-encoder.

