# OpenReview forum: "Unifying Text Semantics and Graph Structures for Temporal Text-attributed Graphs with Large Language Models"
_NeurIPS.cc/2025/Conference — NeurIPS 2025 poster_

### Official Review · Reviewer_C47o · 2025-06-12

**Clarity:** 3
**Significance:** 3
**Originality:** 3
**Rating:** 5
**Confidence:** 4

**Summary:**

This paper focuses on representation learning for temporal text-attributed graphs. To address the limitations of existing models in overlooking the temporal evolution of text semantics and the interactions between semantic structure and text, the authors propose a CROSS framework that can seamlessly extend existing TGNNs for TTAG modeling. The proposed framework consists of a temporal semantics extraction component that uses LLM to summarize the historical semantics of each node, and the Cross-modal Mixer layer to perform information interaction between the node's evolving semantics and structures. Extensive experimental results on real-world datasets verify its superiority.

**Questions:**

1. For equation 1, the interaction frequency between entities can vary significantly across different time periods, being dense in some intervals and sparse in others. Does the use of a fixed sampling interval risk underrepresenting historical information during sparse periods? How does this strategy compare to an alternative that samples the most recent m interactions instead?

2. While the semantic layer effectively captures the historical semantic information of nodes, why does the structural layer not incorporate the historical structural context of the nodes? I suggest the authors clarify this design choice and discuss whether integrating historical structural information into the structural layer could further improve the model’s performance.

**Ethical Concerns:**

["NO or VERY MINOR ethics concerns only"]

**Final Justification:**

Accept

**Limitations:**

Yes

**Quality:**

3

**Strengths And Weaknesses:**

Strength:

1. The paper is clearly written and easy to follow.

2. The technical design of the proposed model is well-justified and thoughtfully developed.

3. Extensive experiments have been conducted to demonstrate the effectiveness of the proposed framework.

Weakness:

1. In Section 2, Definition 2 lacks rigor. It would be more precise to refer to it as Temporal Text-attributed Graph Representation Learning, as the term “modeling” is ambiguous in this context. Additionally, Definition 3 seems to have limited relevance to the core contributions of the paper.

2. The Related Work section does not adequately discuss recent models that employ LLMs for dynamic graph modeling [1,2,3]. Furthermore, the baseline comparisons should also include models that do not incorporate textual information, as they address similar tasks such as temporal graph node classification and link prediction.

[1] Large Language Models-guided Dynamic Adaptation for Temporal Knowledge Graph Reasoning

[2] Back to the Future: Towards Explainable Temporal Reasoning with Large Language Models

[3] TimeR4: Time-aware Retrieval-Augmented Large Language Models for Temporal Knowledge Graph Question Answering

---

> ### Author Rebuttal · Authors · 2025-07-30
>
> We would like to sincerely thank Reviewer C47o for providing a detailed review with insightful questions! We will revise our paper accordingly.
>
> >**W1:** Explanation of terminology definition.
>
> Thank you for your advice on the rigor of Definition 2. Although some works [1] use "modeling" as a substitute for "representation learning", this indeed lacks precision. To ensure clarity and rigor, we will revise the term "Temporal Text-attributed Graph Modeling" to "Temporal Text-attributed Graph Representation Learning" in our revised manuscripts.
>
> As for Definition 3, we explicitly distinguish between LLMs and LMs to prevent potential misunderstandings, as some literature [2, 3] considers LLMs to be a specific subclass of LMs. However, in our work, we enhance the **LLM** with dynamic reasoning capabilities to generate evolving textual summaries, which are subsequently encoded into d-dimensional vectors by an **LM**. This separation clarifies their respective roles and helps prevent conceptual confusion.
>
> [1] Towards Adaptive Neighborhood for Advancing Temporal Interaction Graph Modeling, KDD 2024.
> [2] A Comprehensive Overview of Large Language Models, ACM TIST, 2023.
> [3] GLBench: A Comprehensive Benchmark for Graph with Large Language Models, NeurIPS 2024.
>
> >**W2-1:** Comparison with temporal knowledge graph reasoning (TKGR).
>
> **Technical Comparison.** In the domain of NLP, several LLM-based TKGR methods [1, 2, 3, 4, 5] have been proposed to enhance the graph reasoning capabilities of LLMs within temporal knowledge graphs (TKGs). We compare these methods with CROSS as follows.
>
> * **Similarity.** Essentially, both works can predict future knowledge based on historical knowledge in dynamic graph-structured data.
> * **Key Differences.** CROSS enhances LLMs for representation learning on TTAGs while LLM-based TKGR methods aim to improve LLMs for graph reasoning over TKGs. Moreover, CROSS extracts structural information via GNN-based aggregation, while LLM-based TKGR methods typically rely on explainable rule-based quadruples [1].
>
> LLM-based TKGR methods cannot be fully applied to TTAGs, as **TTAGs are not necessarily TKGs**. For example, the Googlemap_CT dataset records user–restaurant review interactions rather than structured knowledge triples. Additionally, although LLM-based TKGR methods primarily focus on graph reasoning tasks, CROSS is designed to generate node representations to support various downstream applications. Therefore, these methods, while valuable, **may fall outside the scope of our work**. We will discuss, analyze, and appropriately cite these works in our revised paper.
>
> Key differences are summarized as follows.
>
> ||LLM-based TKGR methods|CROSS|
> |:-|:-|:-|
> |Domain|NLP|ML|
> |Task|Knowledge graph reasoning|Representation learning|
> |Data|TKGs|TTAGs|
> |Role of LLMs|Predictor/Enhancer|Enhancer|
> |Structure Extraction|Rule-based quadruples|GNN-based aggregation|
>
> **Empirical Comparison.** We conduct empirical comparison with a recent LLM-based TKGR method, **LLM-DA**, using TiRGN as its graph-based reasoning module [1]. Other details are available in its original publication. Experiments are conducted on the ICEWS1819 dataset, as it can be structured as a temporal knowledge graph. The results are presented in Tab. 1.
>
> From the results, we observe that **CROSS consistently outperforms LLM-DA**, highlighting its effectiveness once again.
>
> Table 1: MRR result (%) for temporal link prediction on ICEWS1819.
>
> | |Transductive|Inductive|
> |:-:|:-:|:-:|
> |LLM-DA|76.42|57.43|
> |CROSS|**95.77**|**87.80**|
>
> [1] Large Language Models-guided Dynamic Adaptation for Temporal Knowledge Graph Reasoning, NeurIPS 2024.
> [2] Back to the Future: Towards Explainable Temporal Reasoning with Large Language Models, WWW 2024.
> [3] TimeR4: Time-aware Retrieval-Augmented Large Language Models for Temporal Knowledge Graph Question Answering, EMNLP 2024.
> [4] Chain of History: Learning and Forecasting with LLMs for Temporal Knowledge Graph Completion, ACL 2023.
> [5] ChatRule: Mining Logical Rules with Large Language Models for Knowledge Graph Reasoning, PAKDD 2025.
>
>
> >**W2-2:** Baseline clarification.
>
> We would like to clarify that **our baselines already cover both types of models**: (1) those that explicitly incorporate textual information (e.g., LKD4DyTAG), and (2) those that do not (e.g., FreeDyG, DyGFormer). Under this comprehensive comparison, CROSS consistently achieves the best performance, demonstrating its effectiveness.
>
> We will highlight this more prominently in our revised version.
>
> >**Q1:** Sampling of LLM reasoning timestamp.
>
> We would like to clarify that **CROSS samples LLM reasoning timestamps based on fixed interaction intervals, rather than fixed time intervals.**
>
> **Motivation Analysis.** As noted in Line 138, for each node, we first derive its interaction timestamps and then sample timestamps for LLM reasoning **at fixed interaction intervals**. As discussed in Line 135, this strategy ensures that each LLM reasoning step accesses a balanced subset of interactions, thereby mitigating the issue of uneven interaction density across different time periods.
>
> **Empirical Evaluation.** To empirically validate the above analysis, we conduct an ablation study with two additional variants:
>
> * **w/ FTI (Fixed Time Interval)**: For each node, we compute its interaction time span and uniformly sample $m$ timestamps to enable the LLM to summarize at fixed time intervals.
> * **w/ RN (Recent Neighbors)**: For each node at each timestamp, we directly instruct the LLM to summarize the $b$ most recent neighbors.
>
> The corresponding results are reported in Tab. 2. From the results, we find that:
>
> * **Summarizing at fixed time intervals leads to inferior performance.** This confirms the presence of uneven interaction density across different time periods. CROSS addresses this issue by sampling at fixed interaction intervals, ensuring balanced input interactions at each LLM reasoning step.
> * **Summarizing recent interactions also yields suboptimal results.** This is likely due to its narrow focus on short-term contexts, failing to capture the long-range semantic dynamics crucial for TTAG modeling.
>
> Table 2: MRR results (%) for temporal link prediction on Enron under $b=20$.
>
> ||TGAT|TGN|
> |:-:|:-:|:-:|
> |w/ FTI (Trans)|82.47|85.20|
> |w/ RN (Trans)|78.46|83.38|
> |CROSS (Trans)|**95.58**|**95.84**|
> |w/ FTI (ind)|70.21|71.77|
> |w/ RN (ind)|65.60|68.87
> |CROSS (Ind)|**81.52**|**82.38**|
>
> >**Q2:** Motivation of Co-encoder.
>
> As stated in Line 189 and illustrated in Fig. 2 (Line 132), we clarify that **the structural layer already incorporates historical structural information via TGNN-based aggregation.**
>
> **Motivation Analysis.** In CROSS, the semantic layer effectively captures semantic information, while the structural layer fully models structural contexts. As mentioned in the Introduction, the fundamental motivation of our Co-encoder is the assumption that **semantics and structures can reinforce each other.** Therefore, **unification should occur throughout the entire encoding process**, rather than shallow fusion solely at input or output. To achieve this, we propose to **iteratively and hierarchically** unify these two modalities during encoding, enabling their mutual reinforcement and producing cohesive representations for effective TTAG modeling (Lines 181–185, 206–209, and 223–224).
>
> **Empirical Evaluation.** Our ablation study (Tab. 2, Line 275) provides empirical support for the encoder design. Specifically, we conduct experiments to evaluate different design choices of the encoder, including semantic-only encoding, structural-only encoding, and the proposed co-encoding architecture. Under such extensive comparison, **CROSS consistently outperforms all variants**, demonstrating the effectiveness and necessity of the Co-encoder design.
>
> Once again, thank you for your valuable review. You truly have helped us to improve our work!

---

> > ### Comment · Reviewer_C47o · 2025-08-07
> >
> > Thanks for your reply. Your response has addressed my concerns, and I will keep my positive score.

---

> ### Author Response · Authors · 2025-08-07
> **Thanks for your positive assessment!**
>
> Thank you very much for your positive assessment! Your insightful comments have been truly valuable to our work, and we will revise the paper accordingly.
>
> Once again, we sincerely appreciate your thoughtful feedback and support.

---

### Official Review · Reviewer_db5y · 2025-06-30

**Clarity:** 3
**Significance:** 2
**Originality:** 2
**Rating:** 5
**Confidence:** 4

**Summary:**

This paper focuses on learning temporal text-attributed graphs (TTAGs) with LLMs. The proposed method first extracts temporal semantics using LLMs and introduces a modal-cohesive co-encoding architecture to integrate semantic information and structural information (generated by existing TGNNs). The experiments demonstrate the superiority of the proposed method.

**Questions:**

Q1. Does the prompt need to be adapted according to the datasets?

Q2. Why does the cross-modal mixer use only the last semantic embedding as input, while the cohesive representations (Section 3.3) incorporate all $k$ historical semantic embeddings as input? And why is the cross-model mixer not applied only to the output of the last layer?

**Ethical Concerns:**

["NO or VERY MINOR ethics concerns only"]

**Final Justification:**

This paper proposes a novel and effective approach to TTAG learning. While I noted its limitation in generalizing to broader tasks through the zero-shot capabilities of LLMs, I believe it provides a strong foundation for future work in this area. Therefore, I recommend accepting the paper.

**Limitations:**

L1. The applications of the proposed method are limited to node classification and link prediction, even with the inclusion of LLMs.

**Paper Formatting Concerns:**

No.

**Quality:**

3

**Strengths And Weaknesses:**

S1. The framework is simple and effective. The experiments are well-conducted and demonstrate the superiority of the proposed method.

S2. The writing is good and easy to follow.

W1. The method is relatively straightforward and lacks significant technical innovation. The graph temporal attention layer and transformer encoder are directly borrowed from existing methods without modification. The only novel aspect is the use of LLMs to generate semantic texts, thereby amplifying the semantic information.

W2. However, despite incorporating LLMs, the method does not fully capitalize on their potential. It lacks the ability to generalize to different tasks through zero-shot learning and also cannot be applied to edge classification or text generation.

W3. The overall computational cost is substantial, considering both the calls to LLMs and inference on a traditional DGNN. Furthermore, the temporal graph attention layer incurs a significant computational overhead when there are two layers, as demonstrated in previous studies [R1, R2] (see the costs of TGAT). This issue becomes even more pronounced when there are many candidate destinations, such as one positive destination and 100 negative destinations, as in the case of evaluating the MRR metric.

[R1] Do We Really Need Complicated Model Architectures For Temporal Networks?

[R2] TGB-Seq Benchmark: Challenging Temporal GNNs with Complex Sequential Dynamics

---

> ### Author Rebuttal · Authors · 2025-07-30
>
> We would like to sincerely thank Reviewer db5y for providing a detailed review with insightful questions.
>
> >**W1:** Comparison with existing works.
>
> Unlike existing works in both LLM4Graph and TGNN domains, we propose **a concise yet effective framework for LLM4TTAG**, which fundamentally opens new avenues for future research in TTAG modeling.
>
> To improve readability, we first summarize the key differences between existing works and CROSS, with detailed explanations in the subsequent discussion.
>
> ||LLM4Graph Domain|TGNN Domain|CROSS|
> |:-|:-|:-|:-|
> |Structural Dynamics|✘|✔|✔|
> |Semantic Dynamics|✘|✘|✔|
> |**LLM Dynamic Reasoning**|✘|No LLM|✔|
> |**Modality Fusion Strategy**|shallow at output (after encoding)|shallow at input (before encoding)|**hierarchical (during encoding)**|
>
> **Problem Novelty.**
>
> We address a largely underexplored yet practically important problem of TTAG modeling, with limited methods in this area.
>
> **Methodological Novelty.**
>
> We emphasize the novelty of CROSS by addressing the **fundamental limitations** inherent to both the LLM4Graph and TGNN domains:
>
> * **LLM4Graph domain: LLMs in existing LLM4Graph methods lack dynamic reasoning capabilities and typically adopt shallow output modality fusion after encoding.**
> 	* Most LLMs are pretrained on static corpora without temporal consideration, which inherently limits their capacity for dynamic reasoning. Existing LLM4Graph methods rely on static graphs and still integrate these LLMs through static, one-off reasoning for each node, rendering them ill-suited to capture the temporal dynamics within TTAGs (Line 61).
> 	* Furthermore, these methods integrate structural and semantic representations **only at the output stage**, resulting in shallow modality fusion **after encoding** (Line 782). This limits the model’s ability to capture the intricate interplay between structures and semantics, ultimately leading to suboptimal representations for downstream tasks.
> * **TGNN domain: Existing TGNNs cannot effectively capture the semantic dynamics of TTAGs and often employ shallow input modality fusion before encoding.**
> 	* Existing TGNNs rely heavily on their structural encoders to capture the dynamics of graph structures, overlooking the temporal evolution of text semantics within TTAGs (Line 39). As a result, their learned representations tend to be biased and overly reliant on structural information.
> 	* Additionally, they typically pre-embed textual information as input semantic features and incorporate them **only at the input stage**, leading to shallow modality fusion **before encoding** (Line 45). This early fusion fails to enable mutual reinforcement between structures and semantics, thereby neglecting their interplay for TTAG modeling.
>
> To tackle the above limitations, in this paper, we propose CROSS, which (1) enhances LLMs with dynamic reasoning capabilities to capture semantic dynamics; and (2) facilitates hierarchical modality fusion throughout the entire encoding process. Specifically:
>
> * **We introduce a temporal-aware prompting paradigm to empower LLMs with dynamic reasoning capabilities for extracting semantic dynamics.**
> Unlike statically employing LLMs, as discussed in Line 125, we propose a novel temporal reasoning chain paradigm to advance LLMs with dynamic semantic reasoning capabilities for TTAG modeling. It adopts a recurrent prompting paradigm to dynamically summarize the evolving neighborhoods of nodes, effectively extracting semantic dynamics within TTAGs.
> * **We propose a cohesive architecture to facilitate hierarchical modality fusion throughout the entire encoding process.** Unlike shallow fusion solely at input or output, as stated in Line 158, we design a novel co-encoding architecture that encourages information propagation **at the layer-encoding stage**, enabling hierarchical modality fusion **during encoding**. It facilitates deep cross-modal unification and allows both modalities to mutually reinforce each other, leading to cohesive representations for TTAG modeling.
>
> >**W2, L1:** Zero-shot and extension to edge classification.
>
> Our focus is on TTAG representation learning. Therefore, we argue that text generation falls outside the scope of this work. Nevertheless, CROSS can be easily extended to edge classification. For completeness, we evaluate two edge classification settings:
>
> * **Supervised setting**: Models are trained directly using edge labels.
> * **Zero-shot setting**: Models are first trained on temporal link prediction, and a 2-layer MLP is subsequently fine-tuned on top of the frozen encoder for edge classification.
>
> Both implementation and model configurations follow DTGB, with DyGFormer adopted as the backbone. The results on Enron are reported in Tabs. 1 & 2. From these results, we observe that:
>
> * **CROSS still achieves the best performance in both supervised and zero-shot edge classification**, proving its strong generalization and transferability across tasks.
> * All models exhibit a performance drop in the zero-shot setting. This highlights the presence of a transfer learning challenge in TTAG modeling, which warrants future research.
>
> Table 1: Precision, Recall, F1 (%) results for supervised edge classification.
>
> | |JODIE|DyRep|GraphMixer|DyGFormer|CROSS|
> |:-:|:-:|:-:|:-:|:-:|:-:|
> |Precision|65.68|66.25|63.13|66.01|**78.62**|
> |Recall|64.72|63.90|57.35|58.06|**72.74**|
> |F1|64.78|64.32|55.07|56.04|**70.91**|
>
> Table 2: Precision, Recall, F1 (%) results for zero-shot edge classification.
>
> | |JODIE|DyRep|GraphMixer|DyGFormer|CROSS|
> |:-:|:-:|:-:|:-:|:-:|:-:|
> |Precision|47.52|50.28|44.28|52.82|**65.28**|
> |Recall|40.25|46.91|42.82|47.36|**60.22**|
> |F1|42.88|44.58|43.66|45.20|**58.29**|
>
> >**W3:** Costs of LLM calls and model training.
>
> We analyze the cost of LLM calls in **Sec. G (Line 806)** and provide an empirical efficiency study of model training in **Sec. H (Line 835)**.
>
> * **Despite involving LLM calls, CROSS remains applicable for large-scale TTAGs.** We report the token cost and time consumption of LLM calls in Tab. 7 (Line 817). From these results, we find that CROSS maintains acceptable LLM costs even on our industrial dataset, demonstrating its scalability for large-scale TTAGs.
> * **During training, CROSS exhibits no significant extra overhead compared to its backbone, as they share the same number of encoding layers.** We report the training times of models in Fig. 13 (Line 841). The cost of CROSS does not increase significantly compared to its backbone. This is because we strictly follow the hyper-parameters of DyGLib during training (Line 750), ensuring the same number of layers between CROSS and its backbone.
> * **CROSS can adjust the number of encoding layers to fit different computational resources while consistently achieving the best results.** As shown in Fig. 6 (Line 314), CROSS demonstrates remarkable robustness to the number of layers. This indicates that CROSS can flexibly adapt its encoding layers as needed while achieving the best performance.
>
> We acknowledge that computing MRR is relatively more time-consuming than AUC. However, this overhead is not introduced by CROSS itself. We adopt MRR because it is a discriminative and increasingly popular metric in temporal graph learning, as detailed in W5 to Reviewer B85N. We also report other metrics with AUC and Hits@10 in Tab. 6 (Line 797).
>
> >**Q1:** Prompt adaptation for different datasets.
>
> As detailed in **Lines 801–805**, we have carefully considered prompt adaptation across datasets. **Only minor keyword adjustments are required** for the prompts when applied to different datasets. For instance, in the Enron dataset, the terms “item” and “review” are replaced with “user” and “email”, respectively. Further details of prompt design can be found in Sec. F (Line 798).
>
> >**Q2:** Motivation of Mixer.
>
> We discuss the motivation of our Mixer in **Lines 213-219**, and all design choices are empirically validated in **ablation study (Line 322)**.
>
> **Motivation Analysis.** As mentioned in the Introduction, the key motivation of our Mixer is the assumption that **semantics and structures can reinforce each other.** Consequently, **unification should occur throughout the entire encoding process rather than solely after encoding** (Lines 181–185, 206–209, and 223-224). Moreover, as discussed in Lines 216–219, the Mixer integrates only the last semantic representation based on three considerations: **efficiency, temporal awareness, and empirical evidence** where mixing all semantic representations leads to suboptimal performance.
>
> **Empirical Evaluation.** Our ablation study (Line 322) provides empirical support for the Mixer's design:
>
> * **Mixing at output leads to suboptimal performance.** The w/o CM variant (mix at output) exhibits performance degradation. This suggests that fusing bi-modal information only after encoding fails to exploit their mutual reinforcement, resulting in suboptimal performance.
> * **Mixing all semantic representations also leads to suboptimal performance.** The w/ CM$_{\text{all}}$ variant (mix all semantics) also exhibits performance degradation. This is because aggregating all semantic representations into a single vector will lead to uniform attention scores in the next Transformer layers (since this vector closely resembles other semantic representations), making representations indistinguishable and ultimately degrading their quality.
>
> Once again, thank you for your invaluable suggestions. Your questions were truly helpful, and we hope our responses have addressed your concerns satisfactorily.

---

> > ### Comment · Reviewer_db5y · 2025-08-01
> >
> > Thank you for your reply. Your response has addressed my concerns, and I believe this work will inspire future work in this area. As such, I am happy to raise my score.

---

> ### Author Response · Authors · 2025-08-01
> **Thanks for your positive feedback!**
>
> We are sincerely grateful for your positive assessment and for raising your score! We truly appreciate your thoughtful recognition of our rebuttal, particularly regarding the novelty and scalability of CROSS. We will revise our paper based on your reviews.
>
> Once again, thank you for your valuable insights. Your detailed comments have been immensely helpful to us!

---

### Official Review · Reviewer_B85N · 2025-06-30

**Clarity:** 3
**Significance:** 3
**Originality:** 3
**Rating:** 3
**Confidence:** 4

**Summary:**

CROSS (Cohesive Representations Of Semantics and Structures) is a framework designed for modeling Temporal Textual Attribute Graphs (TTAGs). TTAGs combine dynamic textual semantics with evolving graph structures, introducing significant complexity. The study highlights that existing Temporal Graph Neural Networks (TGNNs) do not adequately consider the textual features of dynamic networks.
To address this, CROSS decomposes the TTAG modeling process into two stages:
- Temporal Semantic Extraction: Leveraging large language models (LLMs), language models (LMs), and time encoding to capture temporal semantics in text.
- Semantic-Structural Information Integration: Using a Cross-modal Mixer to automatically transform and fuse bimodal information at hierarchical granularities.

**Questions:**

Questions:
1. Eq. 1 uses n/m as the sampling step. How does the model perform when only sample the nearest m neighbors?
2. In Line 174, do the k timestamps refer to the previously sampled m timestamps, or to k neighbors (i.e., not sampled timestamps)?

Other questions please refer to Weaknesses.

**Ethical Concerns:**

["NO or VERY MINOR ethics concerns only"]

**Limitations:**

yes

**Quality:**

3

**Strengths And Weaknesses:**

Strengths:

- The paper addresses a very novel and timely problem, with limited existing work in this area.
- The writing is clear and easy to follow.
- The experiments are comprehensive and well-executed.

Weaknesses:
1. In Eq. 2, m timestamps are sampled for each node u, and for each timestamp, all historical neighbors before that time are retrieved. This implies that the LLM-based summarization process needs to be performed n × m times. Is this computational cost too high?
2. For the LLM(·; PROMPT) module, since all neighbor text information is retrieved, the input can become extremely long. How do the authors address this issue?
3. The summarization ability of LLMs typically lies in extracting and rewriting important information from long text. However, in this work, the LLM is neither jointly trained nor specifically fine-tuned to identify task-relevant information. Thus, it is unclear whether the LLM can effectively summarize useful information from all retrieved neighbor texts. The current method simply feeds all neighbor texts into the LLM and applies summarization. Is this approach reasonable? How do the authors ensure that the summarized information is useful to the downstream task?
4. For temporal text information extraction, the paper first uses an LLM to summarize neighbor texts, then applies a Transformer encoder to embed the summaries into d-dimensional semantic features, and finally concatenates time encodings to capture dynamic patterns.
However, it is unclear whether LLMs can effectively extract temporal dynamics from text, especially without task-specific training. Given this, is most of the model's ability to capture temporal dynamics actually comes from the time encoding, rather than from the textual summaries themselves?
5. Why is Mean Reciprocal Rank (MRR) used as the evaluation metric for the link prediction task? In most existing temporal link prediction works, AUC/AP is typically adopted. Using MRR makes it difficult to clearly compare the effectiveness of the proposed model with prior methods.
6. What is the purpose of the robustness study? How is it related to the core design of the proposed model? It is unclear how robustness is a key aspect of the method, yet it occupies a significant portion of the experimental section. Could the authors clarify the motivation and relevance of this analysis?
7. The paper claims to be the first work on TTAG modeling; however, reference [1] addresses a highly related problem with a similar approach. The authors should discuss the differences and similarities with [1], and if possible, provide experimental comparisons to clarify the contributions of this work.

[1] Unlocking Multi-Modal Potentials for Dynamic Text-Attributed Graph Representation, KDD 2025

---

> ### Author Rebuttal · Authors · 2025-07-30
>
> Many thanks to Reviewer B85N for the thorough and insightful comments, and we will revise our paper accordingly.
>
> >**W1:** LLM cost.
>
> **Clarification for LLM cost.** We would like to clarify the misunderstanding of the costs for LLMs:
>
> * **For each node, timestamps are sampled over its full interaction timestamps, resulting in only $m$ LLM calls per node instead of $n \times m$.** As stated in Eq. 1 (Line 138), we sample $m$ timestamps from the complete $n$ interaction timestamps for each node. This reduces the number of LLM calls per node from $n$ to $m (m \ll n)$.
> * **LLMs are not repeatedly invoked at every timestamp.** As illustrated in the pseudo-code (Line 635), CROSS pre-generates $m$ summaries for each node in advance. During co-encoder training, for node $u$ at time $t$, we only need to derive $u$'s pre-generated summaries that occur before $t$ to construct its representation, thus avoiding redundant LLM calls.
> * **Sampling ensures that the number of LLM calls remains controllable, effectively balancing temporal granularity and cost efficiency.** As shown in Line 133, by restricting the number of LLM calls to $m (m \ll n)$, we avoid the impractical costs from invoking LLMs at every timestamp. We acknowledge that this sampling may unavoidably introduce information loss, but it indeed helps CROSS to achieve the trade-off between temporal granularity and cost efficiency. We also report the results across different sampling intervals in Fig. 14 (Line 851).
>
> **Evaluation for LLM cost.** We also present the empirical cost statistics of LLM calls in Tab. 7 (Line 817). From the results, we observe that **CROSS maintains acceptable LLM costs even on our industrial dataset**, demonstrating its scalability for large-scale TTAGs.
>
> > **W2:** Lengthy prompt.
>
> As illustrated in Fig. 12 (Line 805), historical interactions within the prompt are organized in reverse chronological order. For each prompt, we impose a token budget of 4K tokens. Once the prompt exceeds this limit, **earlier interactions are truncated**.
>
> > **W3:** Rationality of model design.
>
> CROSS consists of two decoupled components with different responsibilities for TTAG modeling:
>
> * **From the data perspective**, the LLM is dynamically enhanced to generate evolving summaries that capture semantic dynamics, which is responsible for enriching the textual modality for the original TTAG data.
> * **From the model perspective**, the Co-encoder hierarchically unifies semantic and structural information to produce representations, which is responsible for specific spatiotemporal modeling and task adaptation.
>
> Based on the above clarification, we argue that the design of CROSS (including LLM usage) is reasonable. This is because:
>
> * **In CROSS, the LLM is designed as an Enhancer for data augmentation, rather than a Predictor that requires fine-tuning for task adaptation.** CROSS dynamically harnesses the LLM to capture the semantic dynamics over time, generating reusable dynamic summaries for semantic data augmentation. This LLM is not designed for label generation or task-specific prediction, making fine-tuning unnecessary.
> * **In CROSS, task adaptation is achieved through the Co-encoder, not the LLM.** Recent studies [1, 2] reveal that directly using LLMs for task prediction exhibits limited effectiveness in spatiotemporal problems. Empirical results from Q2 to Reviewer Fqmi and W2 to Reviewer C47o further confirm these limitations in the context of TTAG modeling. These observations highlight the important role of a Co-encoder in bridging the gap between LLMs and TTAG modeling, enabling the learning of expressive representations for downstream predictions.
> * **Off-the-shelf LLM already yields highly effective summaries that enable CROSS to achieve state-of-the-art performance.** As presented in Tab. 1 (Line 242), off-the-shelf LLMs can already generate sufficient summaries to support the best performance in TTAG modeling. This empirical evidence further validates the rationality of model design.
>
> [1] Are Language Models Actually Useful for Time Series Forecasting?, NeurIPS 2024.
> [2] Can LLMs Understand Time Series Anomalies?, ICLR 2025.
>
> > **W4:** Text-driven dynamics.
>
> **Our prompting paradigm guides LLMs to capture the semantic evolution over time, effectively facilitating text-driven dynamics for TTAG modeling.** This is evident by:
>
> * **Ablation study for text-driven dynamics.** In our ablation study, we evaluate the w/o TRC variant, where reasoning timestamps and their order are shuffled. This prevents the LLM-generated summaries from reflecting the accurate temporal dynamics within TTAGs. As shown in Fig. 7 (Line 330), the w/o TRC leads to **a substantial performance drop**, confirming the presence and significance of text-driven dynamics.
> * **Ablation study for time encoding.** We further evaluate the contribution of time encoding. We introduce an additional variant, w/o TE, where time encoding is directly removed. From the results in Tab. 1, we find that the w/o TE exhibits **only a marginal performance drop**. This indicates that time encoding alone captures limited temporal dynamics for TTAG modeling.
>
> Table 1: MRR results (%) for temporal link prediction using TGN backbone.
>
> ||Enron|GDELT|ICEWS1819|
> |:-:|:-:|:-:|:-:|
> |w/o TE (Trans)|93.62|77.05|91.72|
> |CROSS (Trans)|**95.84**|**77.95**|**94.74**|
> |w/o TE (ind)|80.33|55.31|82.65|
> |CROSS (ind)|**82.38**|**56.65**|**84.01**|
>
> > **W5:** MRR metric.
>
> We report model performance using **AUC and Hits@10 in Tab. 6** (Line 797), and we choose MRR as the main metric because:
>
> * **Similar trends are observed across all metrics**, where CROSS performs best across all datasets and metrics.
> * **MRR has become an increasingly popular metric in temporal graph learning** [1, 2].
> * CROSS boosts many backbones to achieve high scores (> 98%) on metrics like AUC/AP, resulting in similar values across variants. This makes it hard to compare CROSS's performance across different backbones.
>
> [1] TMetaNet: Topological Meta-Learning Framework for Dynamic Link Prediction, ICML 2025.
> [2] TGB 2.0: A Benchmark for Learning on Temporal Knowledge Graphs and Heterogeneous Graphs, NeurIPS 2024.
>
> > **W6:** Clarification of robustness study.
>
> Our robustness study is both well-motivated and reasonable:
>
> * **Motivation.** We clearly state the motivation of our robustness study in **Lines 274–276**: From the results in Tab. 1 (Line 242), we find that CROSS produces similar results across different structural encoding backbones. We hypothesize this stability is due to the robustness of text semantics, and thus design a robustness study to validate this assumption.
> * **Rationality.** We conduct a robustness study under different graph structural environments, including structural perturbations (e.g., injected noise) and structural expansions (e.g., increased encoding layers). This allows us to evaluate model performance in more degraded or complex structural scenarios. The observed robustness of CROSS emphasizes the significance of text semantics in maintaining effectiveness for TTAG modeling.
>
> > **W7:** Comparison with MoMent.
>
> We would like to clarify that we do **not** claim to be the first to explore TTAG modeling. In fact, we explicitly mention a recent TTAG method, LKD4DyTAG (AAAI 2025), and include it in our empirical comparison (Tab. 1, Line 242). Our work presents the first attempt to unify text semantics and graph structures with LLMs for TTAG modeling.
>
> **Technical Comparison.** We first compare CROSS with MoMent [1]:
>
> * **Similarity.** Both works adopt a multi-modal perspective for TTAG modeling.
> * **Key Differences.** CROSS emphasizes modality fusion, facilitating deeper unification of semantic and structural signals. In contrast, MoMent focuses on modality alignment via a pre-existing alignment loss. Additionally, CROSS advances LLMs to explicitly capture temporal dynamics within the semantic modality, while MoMent still relies on static node texts to derive semantic features.
>
> The key differences are summarized below.
>
> ||MoMent|CROSS
> |:-:|:-:|:-:|
> |Modality Handling|Alignment|Fusion
> |LLM Usage|✘|✔
> |Semantic Dynamics|✘|✔
> |Structural Dynamics|✔|✔
>
> **Empirical Comparison.** We notice that MoMent was not accepted by KDD 2025, which remains **an arXiv preprint without official code**. Therefore, we re-implement it based on the details in its manuscript. We use TGN as the backbone.
>
> From the results in Tab. 2, we find that **CROSS consistently outperforms MoMent**, further validating the effectiveness of CROSS.
>
> Table 2: MRR results (%) for temporal link prediction.
>
> ||Enron|GDELT
> |:-:|:-:|:-:|
> |MoMent (Trans)|83.19|88.40
> |CROSS (Trans)|**95.84**|**77.95**
> |MoMent (ind)|67.62|51.50
> |CROSS (ind)|**82.38**|**56.65**
>
> [1] Unlocking Multi-Modal Potentials for Dynamic Text-Attributed Graph Representation, Arxiv 2025.
>
> > **Q1:** Summarizing recent neighbors.
>
> We conduct additional experiments by summarizing the $b$ most recent neighbors for each node at each timestamp, leading to the variant w/ RN. Notably, this significantly increases the number of LLM calls, resulting in substantial costs. As shown in Tab. 3, **simply summarizing recent neighbors results in inferior performance**. This is likely due to the narrow focus on short-term contexts, which fails to capture the long-range semantic dynamics for TTAG modeling.
>
> Table 3: MRR results (%) for temporal link prediction on Enron under $b=20$.
>
> ||TGAT|TGN
> |:-:|:-:|:-:|
> |w/ RN (Trans)|78.46|83.38
> |CROSS (Trans)|**95.58**|**95.84**
> |w/ RN (ind)|65.60|68.87
> |CROSS (Ind)|**81.52**|**82.38**
>
> > **Q2:** Clarification of $k$.
>
> During training, for node $u$ at time $t$, we incorporate all $k$ summaries that occur before $t$ to derive its semantic representation. **This avoids information leakage from future texts.**
>
> Thank you again for your valuable comments. Your feedback truly helps a lot!

---

> > ### Comment · Reviewer_B85N · 2025-08-06
> > **Official Comment by Reviewer B85N**
> >
> > Thank the authors for their feedback. Most of my concerns were addressed, but I still concerns about the evaluation metrics for link prediction. In DTGB, AUC and AP are used for evaluating temporal link prediction performance, and these metrics are also commonly adopted in most link prediction tasks, even recently have some work evaluated in MRR. Additionally, in Table 6, AUC and AP are reported for only two datasets, and the results appear to be inconsistent with those reported in DTGB. The authors should clarify why this discrepancy occurs? I will keep the score.

---

> ### Author Response · Authors · 2025-08-07
> **MRR Clarification and Results for All Datasets**
>
> Thanks for your feedback. We provide further clarification below.
>
> In our preliminary experiments, we evaluate model performance using **five metrics**: AP, AUC, MRR, NDCG@3, and Hits@10. We select MRR as the main metric because it is the most discriminative and increasingly popular in temporal graph learning, as discussed in W5.
>
> We report AUC and Hits@10 results in Tab. 6, omitting some datasets for brevity. **This is because the same trend holds across all datasets**, with CROSS achieving the best performance.
>
> To further confirm this, we retrieve our training logs and compile the complete results for all datasets. From the results, we find that:
> * **CROSS consistently outperforms all baselines across all datasets and metrics.**
> * **Our reported baseline scores are generally higher than those of DTGB.** This is due to the difference in text embedder: DTGB employs early BERT for text embedding, while CROSS adopts the advanced MiniLM (Line 258), which provides greater efficiency and effectiveness.
> * AUC and AP scores tend to be less distinguishable compared to MRR, reinforcing the benefits of MRR.
>
> Once again, thanks for your suggestions. We will include these results in our revised paper.
>
> Table 1: AP results (%) for temporal link prediction.
> ||||Transductive|||||Inductive|||
> |:-:|:-:|:-:|:-:|:-:|:-:|:-:|:-:|:-:|:-:|:-:
> ||Enron|GDELT|ICEWS1819|Googlemap_CT|Industrial|Enron|GDELT|ICEWS1819|Googlemap_CT|Industrial
> |JODIE|97.26±0.2|94.87±0.2|97.62±0.6|84.28±0.1|93.70±0.2|93.91±0.4|91.47±0.2|94.40±1.7|82.33±0.2|82.57±0.5
> |DyRep|95.99±1.3|93.76±0.5|96.04±0.4|77.21±1.2|87.64±0.2|90.99±2.7|91.00±0.7|91.76±0.5|74.44±1.6|77.23±0.4
> |TCL|97.43±0.1|96.16±0.0|99.23±0.0|89.80±0.2|94.33±0.2|93.49±0.4|92.80±0.1|98.17±0.0|88.09±0.0|82.98±0.3
> |CAWN|97.74±0.0|95.50±0.1|98.90±0.0|88.41±0.1|96.37±0.1|94.48±0.2|91.19±0.2|97.31±0.0|86.29±0.0|89.14±0.1
> |PINT|98.28±0.1|96.26±0.2|97.33±0.1|90.20±0.2|96.00±0.2|95.82±0.4|92.28±0.4|98.11±0.1|87.25±0.1|90.91±0.2
> |GraphMixer|96.27±0.2|94.96±0.1|98.60±0.0|80.12±0.3|94.18±0.1|90.40±0.6|90.79±0.1|96.60±0.1|77.43±0.1|82.66±0.2
> |FreeDyG|97.26±0.1|95.02±0.1|99.08±0.1|84.20±0.4|96.22±0.3|92.46±0.8|91.87±0.2|97.29±0.3|79.22±0.1|84.44±0.3
> |LKD4DyTAG|98.42±0.1|96.26±0.3|99.17±0.2|84.88±0.1|97.20±0.1|93.02±0.2|91.19±0.4|98.21±0.0|80.63±0.2|85.88±0.5
> |
> |TGAT|96.94±0.1|95.70±0.1|99.10±0.0|87.11±0.3|93.60±0.6|91.98±0.2|91.57±0.1|97.81±0.0|85.40±0.2|81.25±0.9
> |**CROSS(TGAT)**|**99.67±0.1**|**98.11±0.2**|**99.64±0.1**|**99.97±0.0**|**97.51±0.2**|**97.44±0.2**|**94.38±0.3**|**98.63±0.2**|**97.23±0.1**|**93.41±0.5**
> |
> |TGN|97.98±0.2|95.35±0.3|99.05±0.1|91.52±0.1|95.00±0.6|94.58±0.5|90.23±0.7|97.48±0.1|90.00±0.0|85.48±1.5
> |**CROSS(TGN)**|**99.71±0.1**|**98.07±0.3**|**99.73±0.3**|**99.98±0.0**|**97.90±0.3**|**97.60±0.3**|**93.36±0.7**|**98.74±0.7**|**97.45±0.1**|**94.69±0.1**
> |
> |DyGFormer|97.90±0.2|96.43±0.0|99.17±0.0|81.16±2.1|97.56±0.1|94.99±0.3|93.45±0.1|98.13±0.0|78.74±2.4|89.81±0.1
> |**CROSS(DyGFormer)**|**99.63±0.3**|**98.50±0.3**|**99.78±0.0**|**99.97±0.0**|**98.79±0.1**|**98.25±0.6**|**95.83±0.5**|**99.09±0.0**|**97.25±0.1**|**95.90±0.1**
>
> Table 2: AUC results (%) for temporal link prediction.
> ||||Transductive|||||Inductive|||
> |:-:|:-:|:-:|:-:|:-:|:-:|:-:|:-:|:-:|:-:|:-:
> ||Enron|GDELT|ICEWS1819|Googlemap_CT|Industrial|Enron|GDELT|ICEWS1819|Googlemap_CT|Industrial
> |JODIE|97.50±0.2|95.55±0.2|97.56±0.6|85.25±0.3|94.34±0.2|93.93±0.3|91.41±0.2|93.68±2.0|82.83±0.4|80.55±0.7
> |DyRep|96.56±0.9|94.63±0.2|95.83±0.3|78.02±0.5|87.97±0.3|91.60±2.1|90.71±0.4|90.60±0.5|74.60±1.0|74.25±0.7
> |TCL|97.52±0.2|96.32±0.0|99.18±.01|89.75±0.2|94.97±0.2|93.20±0.4|92.69±0.1|98.04±.02|87.95±0.0|80.58±0.5
> |CAWN|97.76±0.0|95.62±0.0|98.86±.01|88.51±0.1|96.57±0.1|94.07±0.1|90.92±0.1|97.15±.04|86.39±0.1|87.17±0.1
> |PINT|98.00±0.4|96.27±0.2|99.01±0.1|89.36±0.1|98.11±0.3|95.35±0.7|92.52±0.1|98.27±0.5|88.24±0.2|87.25±0.3
> |GraphMixer|96.42±0.2|95.26±0.0|97.27±0.1|80.08±0.3|94.81±0.1|89.99±0.7|90.69±0.1|95.27±0.1|76.57±0.2|80.01±0.4
> |FreeDyG|97.72±0.1|96.32±0.2|99.08±.01|84.37±0.2|95.29±0.1|95.27±0.9|92.23±0.1|97.92±0.1|80.58±0.1|84.29±0.2
> |LKD4DyTAG|97.01±0.1|97.22±0.1|98.00±0.1|83.29±0.1|96.44±0.0|92.82±0.4|93.73±0.1|96.62±0.1|78.76±0.2|82.98±0.1
> |
> |TGAT|97.17±0.1|95.93±0.1|99.05±0.0|87.17±0.4|94.43±0.4|92.09±0.3|91.74±0.1|97.67±0.1|85.33±0.2|78.88±0.6
> |**CROSS(TGAT)**|**99.69±.04**|**98.17±0.2**|**99.62±0.1**|**99.97±0.0**|**97.04±0.2**|**97.53±0.1**|**94.28±0.3**|**98.54±0.3**|**97.20±0.1**|**92.15±0.4**
> |
> |TGN|98.15±0.2|95.67±0.3|99.02±0.1|91.96±0.1|95.49±0.4|94.86±0.5|90.53±0.5|97.47±0.0|90.50±0.0|83.51±1.6
> |**CROSS(TGN)**|**99.70±0.1**|**98.14±0.3**|**99.73±0.3**|**99.97±0.0**|**97.69±0.7**|**97.48±0.4**|**93.69±0.7**|**98.72±0.7**|**97.42±0.0**|**93.17±0.2**
> |
> |DyGFormer|97.78±0.4|96.52±0.0|99.08±.02|80.96±2.2|97.61±0.1|94.24±0.6|93.12±0.1|97.93±0.0|77.99±2.7|87.89±0.2
> |**CROSS(DyGFormer)**|**99.60±0.3**|**98.53±0.3**|**99.75±.03**|**99.97±0.0**|**98.24±0.1**|**98.03±0.7**|**95.57±0.5**|**98.99±0.1**|**97.24±0.1**|**94.70±0.1**

---

### Official Review · Reviewer_Fqmi · 2025-07-01

**Clarity:** 3
**Significance:** 2
**Originality:** 2
**Rating:** 4
**Confidence:** 4

**Summary:**

The authors tackle the under-explored setting of temporal text-attributed graphs (TTAGs), where both the graph topology and the associated node/edge texts evolve over time. They decompose the problem into (i) extracting dynamic text semantics and (ii) unifying those semantics with structural signals. Specifically, they propose CROSS, which first builds a Temporal Semantics Extractor that prompts an LLM at carefully chosen timestamps to summarize a node’s evolving textual neighbourhood, then feeds those summaries—together with structural features—into a Semantic-Structural Co-encoder that mixes the two modalities layer-by-layer via a lightweight cross-modal MLP mixer. Plugged on top of several standard TGNN backbones, CROSS delivers large gains on four public TTAG benchmarks and an industrial e-commerce graph, and it exhibits robustness to structural noise and depth.

**Questions:**

- The co-encoder currently mixes only the latest semantic embedding each layer. Have you explored attention-based fusion over the full history rather than simple exclusion of older summaries?
- Why not compare with strong text-aware TGNNs such as LLM4DyG [1] with dynamic prompting?
- The theoretical section would be more convincing if there is proof that the error introduced by summarising neighbourhood texts instead of feeding raw sequences is bounded. Such analysis or empirical evidence correlating summary quality with link-prediction gains would be beneficial.
- Why are the performances of different models with Semantic Encoding in Table 2 exactly the same?

[1] Llm4dyg: Can large language models solve spatial-temporal problems on dynamic graphs?

**Ethical Concerns:**

["NO or VERY MINOR ethics concerns only"]

**Final Justification:**

All questions and issues have been addressed. I maintain my recommendation.

**Limitations:**

Yes.

**Quality:**

3

**Strengths And Weaknesses:**

## Strengths
- CROSS is conceptually novel and insightful: the framework remains TGNN-agnostic and easy to retrofit to existing models.
- The experimental results are solid: consistently strong empirical gains across backbones, datasets, and transductive/inductive regimes.
- The paper backs up design choices with extensive ablations (including removing components, different LLMs, varying depth, and synthetic noise) and even demonstrates an industrial fraud-detection case, underscoring practical relevance.

## Weaknesses
- The novelty of the proposed CROSS is limited. Temporal Semantics Extractor reduces to periodic LLM-based text summarisation, and Semantic-Structural Mixer is a shallow two-layer MLP. Both mechanisms resemble prior LLM-for-graphs adapters and do not introduce principled advances in temporal reasoning or cross-modal fusion.
- Cross may suffer from complexity and scalability. For web-scale graphs, LLM'based text generation quickly exceeds practical latency and budget limits. The authors report neither empirical runtimes nor FLOPs, and there is no ablation on how accuracy degrades when $m$ or LLM capacity is reduced, so deployability remains unclear.
- Several competing models (e.g., LKD4DyTAG[1]) also support dynamic textual features, but the experiments only feed them static embeddings; hyper-parameter tuning procedures for baselines are also missing. This makes it hard to attribute performance gains to CROSS rather than to sub-optimal baseline configurations.


[1]  Llm-driven knowledge distillation for dynamic text-attributed graphs.

---

> ### Author Rebuttal · Authors · 2025-07-30
>
> Many thanks to Reviewer Fqmi for providing thorough and insightful comments.
>
> >**W1:** Comparison with existing works.
>
> Unlike existing works in both LLM4Graph and TGNN domains, we propose **a concise yet effective framework for LLM4TTAG**, which fundamentally opens new avenues for future research in TTAG modeling.
>
> To improve readability, we first summarize the key differences between existing works and CROSS, with detailed explanations in the subsequent discussion.
>
> ||LLM4Graph Domain|TGNN Domain|CROSS
> |:-|:-|:-|:-
> |Structural Dynamics|✘|✔|✔
> |Semantic Dynamics|✘|✘|✔
> |**LLM Dynamic Reasoning**|✘|No LLM|✔
> |**Modality Fusion Strategy**|shallow at output (after encoding)|shallow at input (before encoding)|**hierarchical (during encoding)**
>
> **Problem Novelty.**
>
> We address a largely underexplored yet practically important problem of TTAG modeling, with limited methods in this area.
>
> **Methodological Novelty.**
>
> We emphasize the novelty of CROSS by addressing the **fundamental limitations** inherent to both the LLM4Graph and TGNN domains:
>
> * **LLM4Graph domain: LLMs in existing LLM4Graph methods lack dynamic reasoning capabilities and typically adopt shallow output modality fusion after encoding.**
> 	* Most LLMs are pretrained on static corpora without temporal consideration, which inherently limits their capacity for dynamic reasoning. Existing LLM4Graph methods rely on static graphs and still integrate these LLMs through static, one-off reasoning for each node, rendering them ill-suited to capture the temporal dynamics within TTAGs (Line 61).
> 	* Furthermore, these methods integrate structural and semantic representations **only at the output stage**, resulting in shallow modality fusion **after encoding** (Line 782). This limits the model’s ability to capture the intricate interplay between structures and semantics, ultimately leading to suboptimal representations for downstream tasks.
> * **TGNN domain: Existing TGNNs cannot effectively capture the semantic dynamics of TTAGs and often employ shallow input modality fusion before encoding.**
> 	* Existing TGNNs rely heavily on their structural encoders to capture the dynamics of graph structures, overlooking the temporal evolution of text semantics within TTAGs (Line 39). As a result, their learned representations tend to be biased and overly reliant on structural information.
> 	* Additionally, they typically pre-embed textual information as input semantic features and incorporate them **only at the input stage**, leading to shallow modality fusion **before encoding** (Line 45). This early fusion fails to enable mutual reinforcement between structures and semantics, thereby neglecting their interplay for TTAG modeling.
>
> To tackle the above limitations, in this paper, we propose CROSS, which (1) enhances LLMs with dynamic reasoning capabilities to capture semantic dynamics; and (2) facilitates hierarchical modality fusion throughout the entire encoding process. Specifically:
>
> * **We introduce a temporal-aware prompting paradigm to empower LLMs with dynamic reasoning capabilities for extracting semantic dynamics.**
> Unlike statically employing LLMs, as discussed in Line 125, we propose a novel temporal reasoning chain paradigm to advance LLMs with dynamic semantic reasoning capabilities for TTAG modeling. It adopts a recurrent prompting paradigm to dynamically summarize the evolving neighborhoods of nodes, effectively extracting semantic dynamics within TTAGs.
> * **We propose a cohesive architecture to facilitate hierarchical modality fusion throughout the entire encoding process.** Unlike shallow fusion solely at input or output, as stated in Line 158, we design a novel co-encoding architecture that encourages information propagation **at the layer-encoding stage**, enabling hierarchical modality fusion **during encoding**. It facilitates deep cross-modal unification and allows both modalities to reinforce each other, leading to cohesive representations for TTAG modeling.
>
> >**W2:** Scalability and ablation.
>
> **Scalability.** We provide the scalability clarification in Sec. G (Line 806) and present an empirical efficiency study in Sec. H (Line 835).
>
> * **From the cost results of LLM calls, we find that CROSS remains applicable for large-scale TTAGs, validating its scalability.** We report the token cost and time consumption of LLM calls in Tab. 7 (Line 817). From these results, we find that CROSS maintains acceptable LLM costs even on our industrial dataset, proving its scalability for large-scale TTAGs.
> * **From the training time results, we find that CROSS exhibits no significant extra overhead compared to its backbone, further validating its scalability.** We also report the training time of CROSS in Fig. 13 (Line 841). From these results, we find that the training cost of CROSS does not increase significantly compared to its backbone, proving its scalability once again.
>
> **Ablation.** We perform an ablation study across different LLMs in Fig. 8 (Line 354) and a parameter study for $m$ in Fig. 14 (Line 852). **Lowering LLM capacity or reducing $m$ will lead to slight performance drops.** Nevertheless, we emphasize that all these variants still outperform baselines, reaffirming the effectiveness of CROSS.
>
> We will revise our manuscript to highlight the above aspects more prominently.
>
> >**W3:** Baseline and hyper-parameter.
>
> **Directly using existing TGNNs to encode dynamic semantic features brings marginal gains or even slight degradation.** In the "Structural Encoding" column of Tab. 2 (Line 275), we report the performance of existing TGNNs when fed with raw static texts or dynamic LLM-generated texts. Under this setting, the performance of LLM-generated texts yields marginal gains or even slight degradation. We attribute this to the excessive irrelevant temporal semantics from high-order relations, which pollutes the representations.
>
> **CROSS and its backbone share the same set of well-tuned hyper-parameters, ensuring a fair comparison.** As noted in Line 750, we strictly follow the hyper-parameters from DyGLib for CROSS and baselines, which have been carefully selected via an exhaustive grid search.
>
> >**Q1:** Motivation of Mixer.
>
> **Motivation Analysis.** As discussed in Lines 216–219, the Mixer integrates only the last semantic representation based on three considerations: **efficiency, temporal awareness, and empirical evidence** where mixing all semantics leads to suboptimal results.
>
> **Empirical Evaluation.** Our ablation study (Line 322) provides empirical support for the Mixer design. From the results of the w/ CM$_{\text{all}}$ variant (mix all semantics), we find that **mixing all semantic representations leads to suboptimal performance.** This is because aggregating all semantic representations into a single vector will lead to uniform attention scores in the next Transformer layer (since this vector closely resembles other semantic representations), making representations indistinguishable and ultimately degrading their quality.
>
> >**Q2:** Comparison with LLM4DyG.
>
> **Technical Comparison.** We first provide a technical comparison between CROSS and LLM4DyG:
>
> * **Similarity.** Both works explore the spatiotemporal reasoning capabilities of LLMs on dynamic graph-structured data.
> * **Key Differences.** CROSS is an LLM-based framework that enhances LLMs' dynamic reasoning capabilities for TTAG representation learning, while LLM4DyG is a benchmark that promotes LLMs' spatiotemporal reasoning capabilities via its designed prompting schemes. Moreover, LLMs in CROSS act as Enhancers for providing dynamic semantics, while in LLM4DyG, they serve directly as Predictors. Additionally, CROSS adopts the time-ordered text stream to describe the graph, whereas LLM4DyG employs triplets with node IDs.
>
> The key differences are outlined below.
>
> ||LLM4DyG|CROSS
> |:-|:-|:-
> |Scope|LLM benchmark with designed prompts|LLM-based framework for representation learning
> |Role of LLMs|Predictor|Enhancer
> |Graph Description in Prompt|Triplets|Time-ordered text stream
>
> **Empirical Comparison.** We also conduct an empirical comparison between CROSS and LLM4DyG. For LLM4DyG, we adopt its prompting strategy (illustrated in Tab. 7 of its publication) to instruct DeepSeek-v2 to perform temporal link prediction. The results are reported in Tab. 1.
>
> From the results, we observe that **CROSS significantly outperforms LLM4DyG**. This demonstrates that LLM-as-Enhancer can better exploit the spatiotemporal modeling capabilities of LLMs for TTAG modeling.
>
> Table 1: F1 result (%) for temporal link prediction.
>
> ||Enron|GDELT|
> |:-:|:-:|:-:|
> |LLM4DyG (Trans)|41.27|58.74|
> |CROSS (Trans)|**86.72**|**75.28**|
> |LLM4DyG (ind)|33.19|49.01|
> |CROSS (ind)|**75.35**|**60.47**|
>
> >**Q3:** Alternative theoretical analysis.
>
> Here, we provide an alternative theoretical analysis to prove the effectiveness of CROSS, showing that unifying text semantics and graph structures instead of solely considering graph structures is bounded.
>
> > **Theorem 1 (Information Bound).** Given the target task $Y$, there exists a constant $\beta\in(0,1]$ such that:
> >
> > $$I(Z_G,Z_T;Y)\geq I(Z_G;Y)+\beta\min\\{H(Y|Z_G),H(Z_T|Z_G)\\}, \tag{1}$$
> >
> > where $Z_G$ denotes the information derived from graph structures, $Z_T$ represents the information captured from text semantics, $I(\cdot|\cdot)$ denotes mutual information, and $H(\cdot|\cdot)$ depicts the condition entropy.
>
> Due to character limitations, we could provide the corresponding proofs at the discussion phase (if permitted). Please feel free to request.
>
> >**Q4:** Clarification of Tab. 2.
>
> As stated in Line 275, the "Semantic Encoding" column of Tab. 2 refers to the setting where **only** the semantic encoder is used, without any structural encodings from TGNNs. Consequently, **the results under this setting are independent of TGNNs**, and thus identical values are reported across them.
>
> Thanks again for your invaluable suggestions! We hope we have addressed your concerns.

---

### Comment · Area_Chair_kcaG · 2025-08-05

Dear reviewers,

Thank you for your effort in the reviews.

As the discussion period ends soon, please read authors' rebuttals and check if they have addressed your concern.

If authors have resolved your questions, do tell them so.

If authors have not resolved your questions, do tell them so too.

Thanks.

AC

---

### Author Response · Authors · 2025-08-06
**Summary of the Rebuttal and Call for Discussion**

Dear reviewers,

We are deeply grateful for your insightful feedback and valuable suggestions. Your comprehensive reviews have guided us in making significant enhancements to our work. Here, we initiate a discussion by summarizing our rebuttals.

In this paper, we address an underexplored problem of TTAG modeling and propose CROSS, a concise yet effective framework for LLM4TTAG that fundamentally opens new avenues in this area. Unlike existing TGNNs that predominantly focus on structural dynamics, we decouple the TTAG modeling process into two phases: (i) temporal semantics extraction, and (ii) semantic–structural information unification. For temporal semantics extraction, beyond statically adopting LLMs in the LLM4Graph domain, we introduce a novel temporal-aware prompting paradigm that advances LLMs with dynamic reasoning capabilities for TTAG modeling. It adopts a recurrent prompting paradigm to dynamically summarize the evolving neighborhoods of nodes, thus extracting semantic dynamics within TTAGs. For information unification, unlike prior methods that perform shallow modality fusion only at input or output, we design a cohesive architecture that enables hierarchical modality fusion throughout the entire encoding process. It allows both modalities to mutually reinforce each other, leading to cohesive representations for TTAG modeling. Extensive experiments on four public datasets and one large-scale industrial dataset show that CROSS achieves state-of-the-art performance and strong robustness.

**The reviewers have generally held positive rating scores based on their initial reviews or after reading our rebuttal.** We would like to express our appreciation for the positive recognition of the strengths of our work, including:

* Novelty and Originality. The studied problem is **"practical and timely with limited methods"** (Reviewers B85N, db5y), and the proposed method is **"novel and thoughtfully developed"** (Reviewers Fqmi, db5y, C47o, B85N).
* Strong Motivation. The overall framework is **"well-motivated"** (Reviewers Fqmi, db5y), and all components are **"well-justified"** (Reviewers Fqmi, db5y, C47o).
* Comprehensive Evaluation. Experimental results are **"extensive and solid"** (Reviewers Fqmi, B85N, C47o) and are **"well-executed"** to show CROSS's effectiveness and robustness (Reviewers B85N, db5y, C47o, Fqmi).
* Clear Presentation. The paper is **"well-written"** (Reviewers B85N, db5y, C47o, Fqmi) and **"easy to follow"** (Reviewers B85N, db5y, C47o, Fqmi).

We have responded individually to each reviewer's questions. To address your concerns and enhance our submission, we have incorporated your suggestions by providing additional clarifications and conducting further experiments. Below is a summary of our rebuttal:

* **Comparison with existing works (Reviewers Fqmi, db5y).** We summarized the limitations of existing works in both TGNN and LLM4Graph domains, and then systematically highlighted the technical differences between CROSS and these methods from the perspectives of LLM utilization and modality fusion. This strongly reaffirms the novelty and contributions of CROSS.
* **Scalability clarification (Reviewers Fqmi, B85N, db5y).** We clarified the misunderstanding about CROSS's scalability from both LLM calls and model training. As illustrated in Sec. G, the LLM cost results in Tab. 7 and the training time results in Fig. 13 confirm that CROSS remains applicable to large-scale TTAGs, validating its scalability.
* **More baselines (Reviewers Fqmi, B85N, C47o).** We further incorporated two additional baselines for comparison, bringing the total to 15 baselines. CROSS still achieves state-of-the-art performance, reaffirming its effectiveness.
* **Ablation for time encoding (Reviewer B85N).** We conducted an additional ablation study on time encoding. Compared to destroying text-driven dynamics, removing time encoding yields only a marginal performance drop, highlighting the importance of text-driven dynamics.
* **Ablation for timestamp sampling (Reviewers B85N, C47o).** We also conducted an additional ablation study by sampling at fixed time intervals or directly summarizing the most recent neighbors with LLMs. The observed performance drop demonstrates the effectiveness of our proposed prompting paradigm.
* **Extension to edge classification (Reviewer db5y).** We extended our evaluation to edge classification in both direct and zero-shot settings. Under such extensive comparisons, CROSS still achieves the best performance.
* Minor modifications. We refined terminology definitions for precision and added a discussion on the temporal knowledge graph reasoning task. We also clarified the motivation and rationality of model design.

Once again, we sincerely thank all reviewers for your valuable feedback and constructive opinions, and we will revise our paper accordingly. If you have any further questions, we are happy to provide additional clarification!

Best,
Submission5864 Authors

---

### Note · Authors · 2025-08-12

Dear reviewers, ACs, SACs, and PCs,

We sincerely appreciate the time, effort, and constructive feedback you have devoted to our submission. Your comments have been invaluable in improving the quality and clarity of our work.

We would like to provide the following remarks for the rebuttal and discussion phases of our submission:

1. **Overall evaluation remains positive.**
	* All reviewers engaged with our rebuttal.
	* The average score across all reviewers is at least **"borderline accept" (≥4.0)**, with a consistent confidence score of 4.
2. **All substantive concerns have been satisfactorily resolved.**
	* Throughout the rebuttal, we carefully addressed each reviewer's concerns point by point, providing detailed clarifications, comparison analyses, and additional experimental results.
	* After reviewing our responses, all reviewers explicitly acknowledged that **their core concerns have been satisfactorily addressed.**
3. **The sole negative score is based on the choice of evaluation metrics, which was thoroughly clarified in our response.**
	* As explained in our response, we choose MRR as the primary metric due to its higher discriminative power and the growing adoption in recent temporal graph learning works [1, 2, 3].
	* While AUC results are already included in Tab. 6 (Line 797), we also provided complete AP and AUC results for all datasets in our response. These metrics consistently validate the effectiveness of our proposed method, and we will include them in our revised paper.

In summary, we believe our submission presents a novel and technically sound contribution to the field of TTAG modeling, and we respectfully hope it will be considered for acceptance at NeurIPS 2025.

Thank you again for your time, thoughtful consideration, and support throughout the reviewing process. We hope these remarks are helpful for the final assessment.

Best regards,
Submission5864 Authors

****
[1] TMetaNet: Topological Meta-Learning Framework for Dynamic Link Prediction, ICML 2025.
[2] TGB 2.0: A Benchmark for Learning on Temporal Knowledge Graphs and Heterogeneous Graphs, NeurIPS 2024.
[3] Temporal Graph Benchmark for Machine Learning on Temporal Graphs, NeurIPS 2023.

---

### Decision · Program_Chairs · 2025-09-17

**Decision:**

Accept (poster)

**Comment:**

In this paper, the authors address the problem of temporal TAG (TTAG) modeling, which is an underexplored topic. They propose the framework of CROSS, leveraging LLM for TTAG modeling. Unlike existing TGNNs that focuses on structural dynamics only, the proposed approach considers temporal semantics, and aims for semantic-structural unification. Experiments on four public datasets and one industrial dataset demonstrate the SOTA performance of CROSS.

Strengths:
- The paper is well motivated, addressing an underexplored yet important problem. The proposed concepts are intuitive and well justified.
- The proposed framework can flexibly work with existing TGNN models, making it widely applicable.
- The experimental results are strong across backbones, datasets, and learning settings.

Weaknesses:
- Complexity and scalability maybe a concern for web-scale graphs.
- Experiments protocol, in particularly, treatment of baselines, can be improved.
- Several pieces of related work should be discussed to better position this paper, and to further clarify and strengthen the technical novelty.

Overall, the paper studies a timely and underexplored topic of TTAG + LLM integration. Key questions have been addressed satisfactory in the rebuttal, and they should be incorporated into the camera ready, if accepted.